# Q-CLIP: Unleashing the Power of Vision-Language Models for Video Quality Assessment through Unified Cross-Modal Adaptation

## Abstract

Accurate and efficient Video Quality Assessment (VQA) has long been a key research challenge. Current mainstream VQA methods typically improve performance by pretraining on large-scale classification datasets (e.g., ImageNet, Kinetics-400), followed by fine-tuning on VQA datasets. However, this strategy presents two significant challenges: (1) merely transferring semantic knowledge learned from pretraining is insufficient for VQA, as video quality depends on multiple factors (e.g., semantics, distortion, motion, aesthetics); (2) pretraining on large-scale datasets demands enormous computational resources, often dozens or even hundreds of times greater than training directly on VQA datasets. Recently, Vision-Language Models (VLMs) have shown remarkable generalization capabilities across a wide range of visual tasks, and have begun to demonstrate promising potential in quality assessment. In this work, we propose Q-CLIP, the first fully VLMs-based framework for VQA. Q-CLIP enhances both visual and textual representations through a Shared Cross-Modal Adapter (SCMA), which contains only a minimal number of trainable parameters and is the only component that requires training. This design significantly reduces computational cost. In addition, we introduce a set of five learnable quality-level prompts to guide the VLMs in perceiving subtle quality variations, thereby further enhancing the model's sensitivity to video quality. Furthermore, we investigate the impact of different frame sampling strategies on VQA performance, and find that frame-difference-based sampling leads to better generalization performance across datasets. Extensive experiments demonstrate that Q-CLIP exhibits excellent performance on several VQA datasets. Code is provided in the supplementary material.

## 1 Introduction

The proliferation of portable filming devices has made video production more accessible, leading to an influx of low-quality videos online. Since video quality directly impacts users' Quality of Experience (QoE), robust Video Quality Assessment (VQA) methods are vital for identifying and filtering subpar content.

Current VQA models fall into two categories: knowledge-driven and data-driven. Knowledge-driven methods (Xu et al., 2014; Saad et al., 2014; Mittal et al., 2015; Korhonen, 2019) depend on hand-crafted features, which often fail to capture the complex factors influencing video quality, resulting in limited reliability. In contrast, data-driven methods (Li et al., 2019; Ying et al., 2021; Li et al., 2022; Sun et al., 2022), enabled by subjective VQA datasets, leverage Deep Neural Networks (DNNs) to learn richer representations and achieve superior performance. Nevertheless, the high cost of subjective annotation constrains dataset scale, hindering the full potential of deep learning in VQA.

To address the data scarcity issue, the mainstream solutions adopt a "pretraining-finetuning" paradigm: models are first pre-trained on large-scale classification datasets (e.g., ImageNet (Deng et al., 2009), Kinetics-400 (Kay et al., 2017)), and then fine-tuned on VQA datasets (Hosu et al., 2017; Ying et al., 2021; Wang et al., 2019). While this approach enhances performance, it introduces two critical limitations. First, classification-based pretraining focuses primarily on semantic

learning, which only partially captures the perceptual aspects of video quality. Studies (Wu et al., 2022; Li et al., 2019; Wang et al., 2021; Mi et al., 2024b; Yuan et al., 2024) have shown that video quality depends on multiple dimensions, including semantics, distortion, motion, aesthetics, etc., many of which are not effectively represented by semantic classification alone. Therefore, semantic knowledge learned from classification tasks is inherently limited in its ability to represent overall video quality. Second, the pretraining stage demands substantially more computational resources, often by orders of magnitude compared to finetuning. Taking FAST-VQA (Wu et al., 2022) as an example, pretraining on Kinetics-400 is roughly 10× and 200× more expensive than finetuning

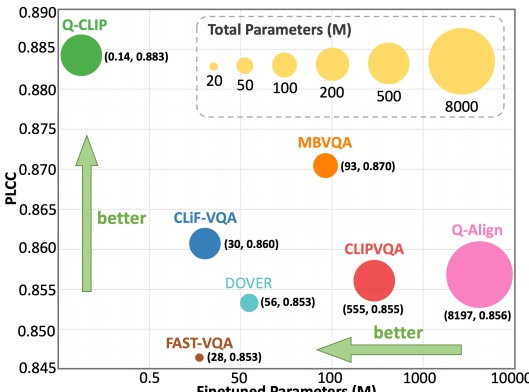

Figure 1: Comparison of Q-CLIP with leading VQA methods on LSVQ.

on LSVQ (Ying et al., 2021) and KoNViD-1k (Hosu et al., 2017), respectively.

Recent advances in Vision-Language Models (VLMs) (Radford et al., 2021; Jia et al., 2021; Zhai et al., 2023; Xu et al., 2024; Bolya et al., 2025) have introduced new perspectives for solving visual tasks. Trained on large-scale image–text pairs, these models acquire rich multimodal knowledge and demonstrate impressive generalization capabilities across domains (Zhang et al., 2024). Unlike traditional classification-based pretraining, which primarily emphasizes semantic discrimination, VLMs inherently encode cross-modal representations that capture the multifaceted nature of video quality, including perceptual distortions (e.g., blur, noise), motion dynamics, aesthetic preferences, and semantic consistency. Moreover, recent studies (Mi et al., 2024b;a; Wang et al., 2023a) demonstrate that VLMs perform well in zero-shot quality prediction across multiple quality-related dimensions, even without task-specific supervision, highlighting their strong potential in quality perception. These properties suggest that VLMs may serve as a promising alternative to classification-based pretraining strategies, offering a more holistic understanding of video quality while also alleviating the computational burden associated with large-scale pretraining.

Despite their strong generalization, efficiently adapting VLMs to VQA remains challenging. The performance of VLMs in downstream tasks is often constrained by limited intra- and cross-modal representational capacity, particularly in fine-grained perceptual tasks (Liang et al., 2022; Qian et al., 2023; Zhang et al., 2023a). However, as a typical fine-grained perceptual task, VQA requires the model to capture subtle quality differences and rely heavily on localized visual cues, which further amplifies the challenge of transferring knowledge from VLMs. Moreover, fine-tuning VLMs for VQA not only incurs substantial computational costs but also risks degrading their original representational capabilities. Given these difficulties, all VQA methods (Mi et al., 2024b; Xing et al., 2024; Yuan et al., 2024) that incorporate VLMs employ VLMs only as auxiliary feature extractors combined with backbone networks, not as standalone frameworks (Appendix. A). Thus, this study aims to explore how to enhance VLMs' perception of quality-related factors while minimizing computational overhead, thereby constructing the first VQA framework entirely based on VLMs.

In addition, fine-grained prompts play a crucial role in providing textual guidance for VLMs (Zhou et al., 2022a;b; Lu et al., 2022; Ju et al., 2022; Yang et al., 2024). Existing VLMs-utilizing quality assessment methods (Wu et al., 2023c; Wang et al., 2023a) often rely on antonym pairs (e.g., "good" and "bad") to guide quality perception. However, such binary prompts provide only coarse-grained supervision and may be insufficient for capturing the full aspects for describing the video quality (Mi et al., 2024b;a). Recent studies (Wu et al., 2024; You et al., 2025) in Large Language Models (LLMs)-based quality assessment show that mapping quality scores to a five-level scale (excellent, good, fair, poor, bad) leads to more accurate predictions. Building on this, we consider similar prompt strategies as a promising direction for enhancing VLMs in VQA.

Based on the above analysis, we introduce Q-CLIP, a VQA method based entirely on VLMs. Specifically, we design a Shared Cross-Modal Adapter (SCMA) to enhance the representations of the visual and textual branches. This adapter consists of only a few fully connected layers **(0.14M)** and is the only component that requires training, significantly reducing computational overhead. As shown in Fig. 1, Q-CLIP achieves the best performance while training only a minimal number of parameters.

In addition, we develop a set of learnable five-level prompts to provide fine-grained textual quality descriptions as input guidance to the VLMs. This allows us to jointly consider the similarity scores between the video and prompts of different quality levels, enabling more accurate quality prediction. Furthermore, we investigate the impact of different frame sampling strategies on VQA performance. Previous works (Wu et al., 2022; Wen et al., 2024) mainly adopt random or uniform sampling, with limited exploration of its impact on VQA performance. Specifically, beyond conventional methods, we explore frame-difference-based sampling strategies to assess its potential benefits for VQA. Our findings offer new insights that may inform and inspire future research in this direction.

Our contributions can be summarized as follows:

- We introduce Q-CLIP, the first VQA model fully based on VLMs. By incorporating an extremely lightweight adapter (SCMA), Q-CLIP effectively boosts the VQA capabilities of VLMs at a remarkably low training cost.
- We design a learnable five-level prompt mechanism to guide VLMs in perceiving subtle quality variations.
- This work presents a systematic study of frame sampling strategies, offering new insights and practical guidance for future research in VQA.
- Extensive experiments demonstrate that Q-CLIP achieves state-of-the-art performance across multiple VQA datasets.

## 2 RELATED WORK

### 2.1 VQA METHODS

**Knowledge-driven.** Knowledge-driven methods (Mittal et al., 2015; 2012; Tu et al., 2021a; Korhonen, 2019; Tu et al., 2021b; Xu et al., 2014; Saad et al., 2014) assess video quality by extracting handcrafted features. For example, VIIDEO (Mittal et al., 2015) utilizes intrinsic statistical regularities of natural videos to capture anomalous information caused by distortion. TLVQM (Korhonen, 2019) extracts low-complexity motion features and high-complexity spatial features. VIDEAL (Tu et al., 2021a) detects and quantifies distortions by extracting a diverse set of perceptual quality features. However, handcrafted features struggle to capture the complex and diverse factors affecting video quality, leading to suboptimal performance.

**Data-driven.** Data-driven methods automatically extract quality-aware features by training DNNs on high-quality VQA datasets. For example, GST-VQA (Chen et al., 2021) and VSFA (Li et al., 2019) use pretrained 2D Convolutional Neural Networks (CNNs) (Simonyan & Zisserman, 2014; He et al., 2016) combined with GRU (Cho et al., 2014) for spatiotemporal modeling, while other studies (Ying et al., 2021; Li et al., 2022; Sun et al., 2022; Wang et al., 2021; Zhang et al., 2023b; Wen et al., 2024) further incorporate 3D-CNNs (Tran et al., 2015; Hara et al., 2018; 2017) to enhance spatiotemporal feature extraction. In addition, Transformer-based VQA (Wu et al., 2023b; 2022; 2023a;e; Liu et al., 2023) is gradually gaining more competitive performance. For example, FAST-VQA (Wu et al., 2022) and FasterVQA (Wu et al., 2023a) sample spatial-temporal grids and utilize modified Video Swin Transformers (Liu et al., 2022). However, these fragment sampling strategies often neglect semantic content. Based on this, DOVER (Wu et al., 2023d) and Zoom-VQA (Zhao et al., 2023) further introduce a semantic branch to enhance FAST-VQA. With the success of VLMs (Radford et al., 2021; Tschannen et al., 2025; Bolya et al., 2025), applying them to VQA has become a growing research focus. CLiF-VQA (Mi et al., 2024b) and PTM-VQA (Yuan et al., 2024) extract human feeling features from CLIP (Radford et al., 2021) under language supervision, serving as a complement to spatiotemporal features. Similarly, MaxVQA (Wu et al., 2023c) and CLIPVQA (Xing et al., 2024) integrate CLIP with backbone networks to extract multimodal features for VQA.

### 2.2 VISION-LANGUAGE MODELS

Vision-Language Models (VLMs) are trained with contrastive learning on large-scale image–text pairs to align visual and textual representations in a shared embedding space. CLIP (Radford et al., 2021) is a representative model known for its robust representations and strong generalization, learned from comparative training on 400 million image-text pairs. Subsequently, a series of

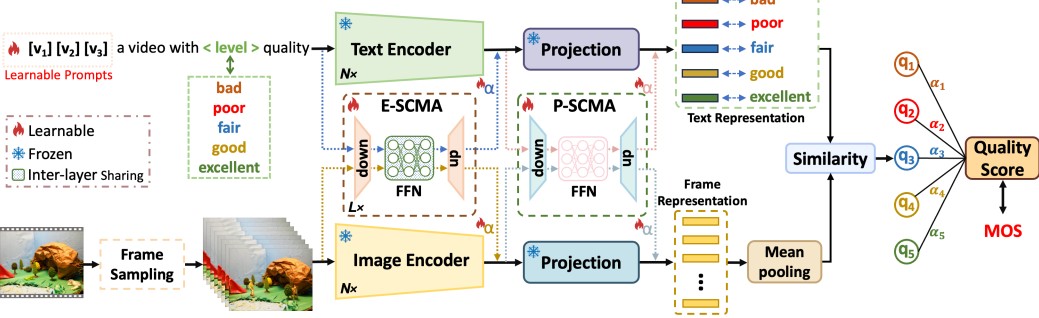

Figure 2: The overall framework of the proposed Q-CLIP.

enhanced versions of CLIP are proposed. For example, MetaCLIP (Xu et al., 2024) enhances the performance of CLIP by curating and balancing the raw data from the network to improve the quality of the training data. SigLIP (Zhai et al., 2023) replaces the original Softmax with Sigmoid when calculating the loss, leading to improved computational efficiency as well as better performance. Furthermore, SigLIP2 (Tschannen et al., 2025) is trained on a larger-scale dataset and unifies previously disjoint training strategies into a structured, multi-stage pipeline, resulting in notable performance improvements. Recently, (Bolya et al., 2025) train CLIP on a larger image dataset and fine-tune it on a 22M-sized video dataset, significantly enhancing its generalization to video data. For example, CoOp (Zhou et al., 2022a), CLIP-Adapter (Gao et al., 2024a), and Tip-Adapter (Zhang et al., 2021) enhance VLMs to enable few-shot image recognition. Moreover, some works extend VLMs to video tasks. VideoCLIP (Xu et al., 2021) replaces image-text pairs with video-text pairs for video understanding. CLIP4Clip (Luo et al., 2022) adapts CLIP for video retrieval by fine-tuning it end-to-end. ActionCLIP (Wang et al., 2023b) and XCLIP (Ni et al., 2022) directly transfer CLIP's visual representations to video recognition.

## 3 PROPOSED METHOD

### 3.1 OVERALL ARCHITECTURE

Our proposed Q-CLIP architecture, illustrated in Fig. 2, is fully built upon the VLMs framework. It enhances the performance of VLMs in VQA by introducing two novel modules: Shared Cross-Modal Adapter (SCMA) and a set of learnable five-level quality prompts. In addition, we investigate the impact of different frame sampling strategies on VQA performance. Beyond conventional random and uniform sampling, we explore a motion-based approach that calculates the difference between each frame and its adjacent frames to measure the intensity of motion. Frames are then sampled according to predefined rules based on these motion differences.

### 3.2 SHARED CROSS-MODAL ADAPTER

The core objective of SCMA is to mitigate feature distribution discrepancies between visual and textual branches, facilitating precise cross-modal alignment. Fig. 3 illustrates the detailed architectural design of SCMA. Specifically, we design two structures of SCMA, E-SCMA and P-SCMA, to align the features of encoder and projection, respectively. By reusing the same SCMA architecture for both branches, the model learns a single, generalized strategy to refine and align features, rather than modality-specific heuristics.

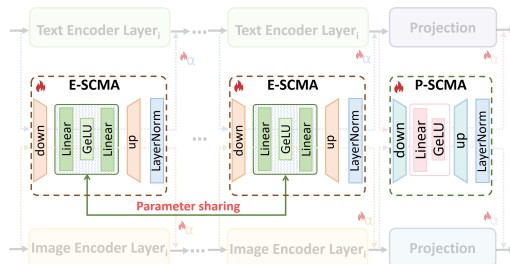

Figure 3: Architecture of the proposed SCMA.

This consistency ensures that visual and textual features are transformed through comparable operations, reducing the risk of divergent optimization directions that could widen the modality gap. However, since VLMs contain multiple encoder layers, adding E-SCMA to each of them would lead to a multiplicative increase in the number of trainable parameters, thereby increasing the risk of model overfitting. To address this issue, we incorporate inter-layer parameter sharing in the Feed Forward Network (FFN) of E-SCMA. This design not only effectively reduces training costs but also helps mitigate the risk of overfitting.

Specifically, we denote the inputs to the visual encoders ($E_k^v$) and text encoders ($E_k^t$) at layer $k$ as $V_k$ and $T_k$, respectively, where $V_k \in \mathbb{R}^{N \times L_v \times D_v}$ ($N$: number of video frames, $L_v$: sequence length of visual tokens, $D_v$: visual hidden dimension), $T_k \in \mathbb{R}^{M \times L_t \times D_t}$ ($M$: number of text instances, $L_t$: sequence length of text tokens, $D_t$: text hidden dimension). The frozen encoder extracts features from the inputs to support subsequent inputs:

$$V_k^{'} = E_k^v(V_k), V_k^{'} \in \mathbb{R}^{N \times L_v \times D_v}; T_k^{'} = E_k^t(T_k), T_k^{'} \in \mathbb{R}^{M \times L_t \times D_t}, \tag{1}$$

For E-SCMA, the processing procedure can be described as follows:

$$\Delta V_k = Up_k(FFN(Down_k(V_k))), \Delta V_k \in \mathbb{R}^{N \times L_v \times D_v}, \tag{2}$$

$$\Delta T_k = Up_k(FFN(Down_k(T_k))), \Delta T_k \in \mathbb{R}^{M \times L_t \times D_t}, \tag{3}$$

where $Up_k$ and $Down_k$ are used for dimension expansion and reduction, respectively. To maintain a consistent feature distribution, LayerNorm is employed for normalization:

$$\Delta V_k^{'} = LayerNorm(\Delta V_k), \Delta T_k^{'} = LayerNorm(\Delta T_k), \tag{4}$$

Subsequently, the output of E-SCMA is combined with the encoder's output to obtain the input for the subsequent module:

$$V_{k+1} = V_k^{'} + \alpha_k \Delta V_k^{'}, V_{k+1} \in \mathbb{R}^{N \times L_v \times D_v}; T_{k+1} = T_k^{'} + \beta_k \Delta T_k^{'}, T_{k+1} \in \mathbb{R}^{M \times L_t \times D_t}, \tag{5}$$

where both $\alpha_k$ and $\beta_k$ are learnable parameters.

For P-SCMA, the processing flow is largely similar to that described above. The only difference lies in FFN, which consists of a single linear layer. This is because the projection module is inherently designed to perform simple linear mappings, facilitating similarity computation between the two modalities. To remain consistent with this lightweight mapping structure, we adopt a more streamlined version.

### 3.3 Learnable Five-level Prompts

Using antonym-based prompts (e.g., good vs. bad) in VLMs has shown promising results for quality perception. However, since quality assessment is a fine-grained prediction task, such binary prompts are overly coarse and may limit the performance of VLMs in capturing subtle quality differences. Fortunately, recent studies on quality perception using LLMs have demonstrated that converting quality scores into discrete quality levels helps models better capture nuanced hierarchical patterns, leading to improved performance. Inspired by this, we argue that a similar design is also beneficial for VLMs. Therefore, we introduce a prompt scheme with five distinct quality levels:

$$p = \text{"a video of"} + <level> + \text{"quality"} \tag{6}$$

Here, $<level>$ represents the five quality levels: excellent, good, fair, poor, bad. However, more specific prompts can often introduce bias into the perception of VLMs. To address this, we introduce learnable prompts to optimize the initial five-level prompt scheme:

$$\hat{p} = Learnable(\text{"X X X"}) + p \tag{7}$$

We initialize the prompts with three "X" tokens and optimize them during training. All other parts of prompts are keep frozen. The prompts used in this work can be formulated as:

$$P = \{\hat{p}_{exc}, \hat{p}_{good}, \hat{p}_{fair}, \hat{p}_{poor}, \hat{p}_{bad}\} \tag{8}$$

### 3.4 Quality Regression

The video and text prompts are processed by the model to obtain video features $V$ and text features $T = \{t_{exc}, t_{good}, t_{fair}, t_{poor}, t_{bad}\}$, respectively. Then, calculate the cosine similarity between the visual content and prompts to predict the score for each dimension:

$$s_k = \frac{t_k \cdot V}{\|t_k\| \|V\|}, k \in \{exc, good, fair, poor, bad\} \tag{9}$$

These similarity scores $S = \{s_{exc}, s_{good}, s_{fair}, s_{poor}, s_{bad}\}$ form a quality-level-related distribution, which is further processed to generate the final quality assessment score. Finally, by applying a weighted sum, the discrete similarity scores are converted into a continuous quality prediction:

$$Q_{pred} = \sum_{k=exc}^{bad} w_k \cdot s_k \tag{10}$$

where $w_k$ are learnable weights adjusting each quality level's contribution to the final prediction.

### 3.5 FRAME-DIFFERENCE-BASED SAMPLING

VQA relies heavily on representative frame samples, as full video sequences are often computationally prohibitive and redundant. While random and uniform sampling are widely used as baselines, they overlook the dynamic characteristics of videos that may correlate with quality perception (e.g., motion intensity). To address this, we systematically investigate the impact of frame sampling strategies on VQA performance, with a particular focus on frame-difference-based sampling, a strategy rarely explored in prior VQA literature. Frame differences, quantified via pixel-wise MSE, reflect motion intensity between consecutive frames:

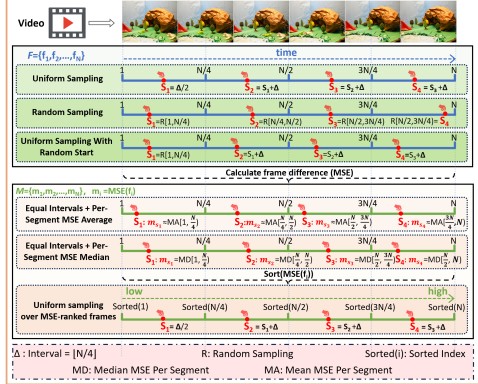

Figure 4: Frame Sampling Diagram.

$$m_t = \frac{1}{2} MSE(v_i, v_{i+1}) + MSE(v_i, v_{i-1})) \quad (11)$$

where $MSE(a, b) = \frac{1}{H \times W \times 3} \sum_p (a_p - b_p)^2$ computes the pixel-wise MSE between frames $a$ and $b$ with ($p$ indexing individual pixels). The first and last frames are compared only with their single adjacent frames. Using the frame difference MSE $\{m_1, m_2, ...m_N\}$, we design three sampling strategies to select frames, as illustrated in Fig. 4, which shows the detailed sampling process. Additional details regarding the sampling process can be found in Appendix. E.

## 4 EXPERIMENTS

### 4.1 EXPERIMENTAL SETUPS

**Datasets.** We verify our model on six datasets: LSVQ (Ying et al., 2021), KoNViD-1k (1200) (Hosu et al., 2017), LIVE-VQC (585) (Sinno & Bovik, 2018), YouTube-UGC (1067) (Wang et al., 2019), CVD2014 (234) (Nuutinen et al., 2016), LIVE-Qualcomm (208) (Ghadiyaram et al., 2017). We pre-train Q-CLIP on LSVQ (28056), with intra-dataset testing on LSVQ$_{test}$ (7400) and LSVQ$_{1080p}$ (3600), and cross-dataset testing on KoNViD-1k and LIVE-VQC. Further, we fine-tune the model on KoNViD-1k, LIVE-VQC, YouTube-UGC, CVD2014 and LIVE-Qualcomm. Following standard practice, we split each dataset into 80% training and 20% testing.

**Evaluation Criteria**. The Spearman Rank Order Correlation Coefficient (SROCC), the Kendall Rank Order Correlation Coefficient (KROCC), the Pearson Linear Correlation Coefficient (PLCC), and the Root Mean Square Error (RMSE) are used as evaluation metrics.

**Implementation Details.** We employ PyTorch framework and an NVIDIA GeForce RTX 4090 card to train the model in all experimental implementations. As most current VLMs are trained on static images, they are not well-equipped to model the temporal dynamics in videos. To address this, we adopt a CLIP variant (Bolya et al., 2025) that has been pre-tuned on video data as our backbone. We sample 8 frames per video as input. We set the initial learning rate to 0.001, the optimizer to AdamW, and use a cosine annealing strategy to dynamically adjust the learning rate. And training is conducted for 8 epochs using a batch size of 12. More experimental details are in Appendix. D.

### 4.2 PRE-TRAINING RESULTS ON LSVQ

We pre-train the proposed Q-CLIP on LSVQ and conduct intra-dataset testing on LSVQ$_{test}$ and LSVQ$_{1080p}$. Additionally, cross-dataset testing performed on KoNViD-1k and LIVE-VQC. Furthermore, we examine the effectiveness of various frame sampling methods. The results are shown in Tab. 1. Frame-difference-based samplings achieve comparable results to traditional methods, such as random and uniform sampling, in intra-dataset testing. In cross-dataset testing, frame-difference-based samplings demonstrate superior performance compared to traditional approaches, indicating better generalization capability. This suggests that frame-difference-based samplings can more effectively select frames that are representative of video quality, particularly in cross-dataset scenarios. In contrast, traditional sampling methods do not consider any intrinsic characteristics of

| Testing Type | | | Intra-dataset Test Datasets | | | | Cross-dataset Test Datasets | | | |
|---|---|---|---|---|---|---|---|---|---|---|
| Testing Datasets | | | **LSVQ$_{test}$** | | **LSVQ$_{1080p}$** | | **KoNViD-1k** | | **LIVE-VQC** | |
| Type | Methods | Source | SROCC↑ | PLCC↑ | SROCC↑ | PLCC↑ | SROCC↑ | PLCC↑ | SROCC↑ | PLCC↑ |
| Knowledge-driven | BRISQUE | *TIP, 2012* | 0.569 | 0.576 | 0.497 | 0.531 | 0.646 | 0.647 | 0.524 | 0.536 |
| | TLVQM | *TIP, 2019* | 0.772 | 0.774 | 0.589 | 0.616 | 0.732 | 0.724 | 0.670 | 0.691 |
| | VIDEVAL | *TIP, 2021* | 0.794 | 0.783 | 0.545 | 0.554 | 0.751 | 0.741 | 0.630 | 0.640 |
| Data-driven | VSFA | *ACMMM, 2019* | 0.801 | 0.796 | 0.675 | 0.704 | 0.784 | 0.794 | 0.734 | 0.772 |
| | PVQ | *CVPR, 2021* | 0.827 | 0.828 | 0.711 | 0.739 | 0.791 | 0.795 | 0.770 | 0.807 |
| | BVQA | *TCSVT, 2022* | 0.852 | 0.854 | 0.771 | 0.782 | 0.834 | 0.837 | 0.816 | 0.824 |
| | FAST-VQA | *ECCV, 2022* | 0.876 | 0.877 | 0.779 | 0.814 | 0.859 | 0.855 | 0.823 | 0.844 |
| | DOVER | *ICCV, 2023* | 0.881 | 0.879 | 0.782 | 0.827 | 0.871 | 0.872 | 0.812 | 0.841 |
| | Zoom-VQA | *CVPR,2023* | 0.886 | 0.879 | 0.799 | 0.819 | 0.877 | 0.875 | 0.814 | 0.833 |
| | MBVQA | *CVPR, 2024* | 0.895 | 0.895 | 0.809 | 0.844 | 0.878 | 0.884 | 0.806 | 0.844 |
| LLMs | Q-Align | *ICML, 2024* | 0.883 | 0.882 | 0.797 | 0.830 | 0.865 | 0.877 | NA | NA |
| VLMs | PTM-VQA | *CVPR, 2024* | 0.855 | 0.864 | 0.736 | 0.782 | 0.824 | 0.830 | 0.785 | 0.737 |
| | CLiF-VQA | *ACMMM, 2024* | 0.886 | 0.887 | 0.790 | 0.832 | 0.877 | 0.874 | **0.834** | 0.855 |
| | CLIPVQA | *TBC, 2025* | 0.881 | 0.883 | 0.782 | 0.827 | 0.864 | 0.887 | 0.781 | **0.871** |
| | **Q-CLIP** -*RandSampl* | | 0.895 | 0.896 | 0.814 | 0.852 | 0.882 | 0.892 | 0.808 | 0.843 |
| | **Q-CLIP** -*UNISampl* | | 0.897 | 0.895 | 0.820 | 0.853 | 0.883 | 0.891 | 0.803 | 0.842 |
| | **Q-CLIP** -*UNIRandStart* | | 0.893 | 0.895 | 0.818 | 0.858 | 0.883 | 0.890 | 0.804 | 0.844 |
| | **Q-CLIP** -*MSESortedUNI* | | 0.891 | 0.893 | 0.812 | 0.852 | 0.888 | 0.894 | 0.810 | 0.845 |
| | **Q-CLIP** -*SegMSEMean* | | 0.897 | 0.896 | 0.820 | 0.855 | 0.889 | 0.895 | 0.813 | 0.851 |
| | **Q-CLIP** -*SegMSEMedian* | | 0.891 | 0.893 | 0.813 | 0.852 | 0.889 | 0.896 | 0.813 | 0.852 |
| | **Q-CLIP** -*Mixed* | | **0.899** | **0.900** | **0.823** | **0.866** | **0.896** | **0.901** | 0.826 | 0.867 |

Table 1: Experimental performance of the pre-trained Q-CLIP on LSVQ on four test sets (LSVQ$_{test}$, LSVQ$_{1080p}$, KoNViD-1k, LIVE-VQC). The best and second-best results are **bolded** and underlined.

| Finetune Datasets | | | **LIVE-VQC** | | **KoNViD-1k** | | **YouTube-UGC** | | **CVD2014** | | **LIVE-Qualcomm** | |
|---|---|---|---|---|---|---|---|---|---|---|---|---|
| Type | Methods | Source | SRCC↑ | PLCC↑ | SRCC↑ | PLCC↑ | SRCC↑ | PLCC↑ | SRCC↑ | PLCC↑ | SRCC↑ | PLCC↑ |
| Knowledge-driven | TLVQM | *TIP, 2019* | 0.799 | 0.803 | 0.773 | 0.768 | 0.669 | 0.659 | 0.830 | 0.850 | 0.770 | 0.810 |
| | VIDEVAL | *TIP, 2021* | 0.752 | 0.751 | 0.783 | 0.780 | 0.779 | 0.773 | NA | NA | NA | NA |
| | RAPIQUE | *OJSP, 2021* | 0.755 | 0.786 | 0.803 | 0.817 | 0.759 | 0.768 | NA | NA | NA | NA |
| Data-driven | VSFA | *ACMMM, 2019* | 0.773 | 0.795 | 0.773 | 0.775 | 0.724 | 0.743 | 0.870 | 0.868 | 0.737 | 0.732 |
| | GST-VQA | *TCSVT, 2021* | NA | NA | 0.814 | 0.825 | NA | NA | 0.831 | 0.844 | 0.801 | 0.825 |
| | PVQ | *CVPR, 2021* | 0.827 | 0.837 | 0.791 | 0.786 | NA | NA | NA | NA | NA | NA |
| | BVQA | *TCSVT, 2022* | 0.841 | 0.839 | 0.835 | 0.834 | 0.825 | 0.818 | 0.863 | 0.883 | 0.833 | 0.837 |
| | FAST-VQA | *ECCV, 2022* | 0.845 | 0.852 | 0.890 | 0.889 | 0.857 | 0.853 | 0.891 | 0.903 | 0.819 | 0.851 |
| | DOVER | *ICCV, 2023* | 0.812 | 0.852 | 0.897 | 0.899 | 0.877 | 0.873 | 0.858 | 0.881 | 0.736 | 0.789 |
| | MBVQA | *CVPR, 2024* | 0.860 | 0.880 | 0.901 | 0.905 | 0.876 | 0.877 | 0.883 | 0.901 | 0.832 | 0.842 |
| VLMs | MaxVQA | *ACMMM, 2023* | 0.854 | 0.873 | 0.894 | 0.895 | 0.894 | 0.890 | NA | NA | NA | NA |
| | PTM-VQA | *CVPR, 2024* | 0.811 | 0.820 | 0.857 | 0.872 | 0.858 | 0.857 | NA | NA | NA | NA |
| | CLiF-VQA | *ACMMM, 2024* | 0.866 | 0.878 | 0.903 | 0.903 | 0.888 | 0.890 | 0.881 | 0.891 | 0.832 | 0.850 |
| | CLIPVQA | *TBC, 2025* | 0.870 | 0.892 | 0.907 | 0.912 | 0.881 | 0.883 | 0.883 | 0.888 | 0.833 | 0.872 |
| | **Q-CLIP** | *Ours* | **0.881** | **0.901** | **0.915** | **0.920** | **0.911** | **0.911** | **0.897** | **0.907** | **0.846** | **0.884** |

Table 2: The finetune results on LIVE-VQC, KoNViD-1k, YouTube-UGC, CVD2014 and LIVE-Qualcomm. The best and second-best results are **bolded** and underlined.

the video frames, which may result in redundant or less informative samples, thereby limiting their representativeness and generalizability. The results indicate that regardless of the sampling strategy employed, Q-CLIP consistently achieves state-of-the-art performance. Moreover, integrating multiple sampling methods during training further enhances the model's overall performance.

Compared with knowledge-driven and data-driven methods, Q-CLIP achieves a significant improvement over all datasets. Furthermore, it demonstrates distinct advantages over Q-Align, which is based on LLMs. Compared with VLMs-utilizing methods, although Q-CLIP performs slightly lower than CLiF-VQA and CLIPVQA on the LIVE-VQC dataset, it significantly outperforms them on the other three datasets. Specifically, Q-CLIP improves over CLiF-VQA and CLIPVQA by up to **3.7%** in SROCC and **2.9%** in PLCC.

### 4.3 FINE-TUNING RESULTS ON SMALL DATASETS

After pre-training on LSVQ, we fine-tune Q-CLIP on five small datasets (LIVE-VQC, KoNViD-1k, YouTube-UGC, CVD2014, LIVE-Qualcomm), as shown in Tab. 2. Specifically, we use uniform sampling as the sampling strategy during fine-tuning. As can be seen, Q-CLIP achieves unprecedented performance on all five datasets. Compared to the current best performance, Q-CLIP improves the average performance on SROCC and PLCC by **1.39%** and **1.22%**, respectively. Furthermore, Q-CLIP outperforms the state-of-the-art VLMs-utilizing method CLIPVQA by **1.74%** and **1.72%** in SROCC and PLCC, respectively. The results further illustrate the validity of Q-CLIP.

### 4.4 COMPARISON WITH OTHER FINE-TUNING METHODS

To validate the effectiveness of SCMA, we compare it against several mainstream fine-tuning

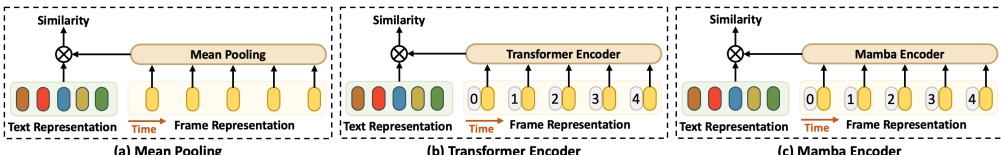

Figure 5: Ablation on the number of E-SCMA Layers.

Figure 6: Ablation on Frame Feature Fusion.

approaches, including full fine-tuning, CoOp (Zhou et al., 2022c), VPT (Jia et al., 2022), CLIP-Adapter (Gao et al., 2024b), and LoRA (Hu et al., 2022), as shown in Tab. 3. Full fine-tuning of CLIP does not yield satisfactory performance. This is primarily because updating all parameters not only disrupts the well-learned representations of CLIP, but also suffers from insufficient training data, making it difficult to effectively optimize the entire model. Due to CoOp optimizing only the prompt and VPT optimizing only the visual branch, both methods perform poorly. In contrast, CLIP-Adapter and LoRA yield more competitive results by preserving the pre-trained knowledge of CLIP and enabling effective adaptation. Nevertheless, their performance still falls short of our proposed SCMA, which demonstrates superior effectiveness in the VQA task. Additional analysis is provided in the Appendix. C.

| Methods | SROCC | PLCC |
|---|---|---|
| Full fine-tuning | 0.816 | 0.811 |
| CoOp | 0.763 | 0.764 |
| VPT | 0.823 | 0.820 |
| CLIP-Adapter | 0.881 | 0.884 |
| LoRA | 0.883 | 0.883 |
| Ours | **0.897** | **0.895** |

Table 3: Comparison of different fine-tuning methods on LSVQ.

## 4.5 ABLATION STUDIES

We conduct experimental analysis to evaluate the effectiveness of each component. Ablation experiments are by default based on a uniform sampling strategy. See Appendix. F for more ablat

**Ablation on SCMA.** We validate the effectiveness of SCMA on LSVQ, as shown in Tab. 4. Applying SCMA to either the visual or textual branch individually results in limited performance. When SCMA is applied to both branches without parameter sharing, the performance improves notably. Sharing SCMA across the two branches leads to further gains, and the best results are achieved when inter-layer sharing is additionally introduced. These results validate the effectiveness of our proposed SCMA architecture, which jointly leverages branch-wise and inter-layer sharing.

| Visual | Text | Sharing | Layer sharing | SROCC | PLCC |
|---|---|---|---|---|---|
| ✓ | | | | 0.866 | 0.864 |
| | ✓ | | | 0.837 | 0.838 |
| ✓ | ✓ | | | 0.875 | 0.878 |
| ✓ | ✓ | ✓ | | 0.885 | 0.886 |
| ✓ | ✓ | ✓ | ✓ | **0.895** | **0.897** |

Table 4: Ablation on SCMA.

Furthermore, since VLMs typically consist of multiple layers, we further investigate the impact of the number of inserted E-SCMA layers on model performance. As shown in Fig. 5, we train the model on $LSVQ_{train}$ and evaluate it on $LSVQ_{test}$, $LSVQ_{1080p}$, KoNViD-1k, and LIVE-VQC. The results demonstrate that as the number of E-SCMA layers increases, the model performance consistently improves. Notably, all comparative experiments in this paper are based on a 6-layer E-SCMA configuration, suggesting that further performance gains can be achieved by increasing the number of E-SCMA layers.

**Ablation on Prompts.** Most existing VLMs-utilizing quality assessment methods (Wu et al., 2023c; Wang et al., 2023a) utilize antonym-based prompts. To validate the effectiveness of our proposed prompts, we compare it against the antonym-based prompts, as shown in Tab. 5. Compared to antonym-based prompts, our five-level prompt design offers a clear advantage. Furthermore, performance is further improved by introducing learnable parameters.

| Datasets | $LSVQ_{test}$ | | KoNViD-1k | | LIVE-VQC | |
|---|---|---|---|---|---|---|
| Prompts | SROCC | PLCC | SROCC | PLCC | SROCC | PLCC |
| *Antonym* | 0.883 | 0.881 | 0.866 | 0.867 | 0.791 | 0.820 |
| *Antonym\** | 0.885 | 0.883 | 0.871 | 0.879 | 0.789 | 0.826 |
| *Five levels* | 0.891 | 0.891 | 0.874 | 0.885 | 0.798 | 0.837 |
| *Five levels\** | **0.895** | **0.897** | **0.883** | **0.891** | **0.803** | **0.842** |

Table 5: Ablation on prompts. \* : learnable.

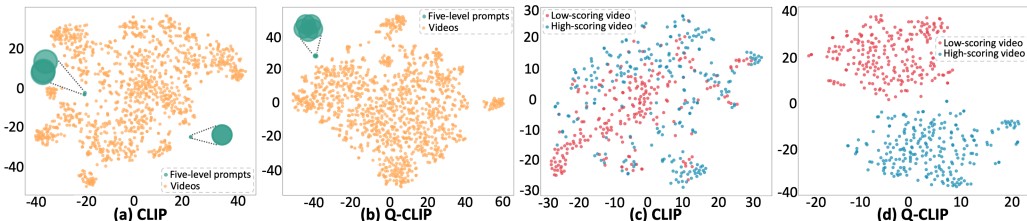

Figure 8: t-SNE visualizations on KoNViD-1k dataset.

**Ablation on Frame Feature Fusion.** As shown in Fig. 6, we compare three frame feature fusion strategies: Transformer (Vaswani et al., 2017), Mamba (Gu & Dao, 2023), and mean pooling. The results, reported in Fig. 7, reveal that mean pooling surprisingly outperforms both Transformer and Mamba. Although video frames inherently contain temporal information, this outcome can be explained by the fact that the CLIP backbone we employ has been pre-finetuned on video datasets with mean pooling. Maintaining this setting is therefore more consistent with the backbone's learned representations, whereas introducing additional temporal modeling may disturb the pre-trained features and lead to suboptimal performance.

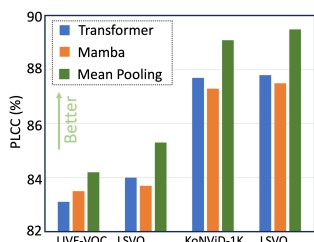

Figure 7: Comparison of different fusion strategies on LSVQ.

### 4.6 EFFICIENCY

Compared to models specifically designed for VQA, VLMs typically have a larger number of parameters, making model efficiency a critical consideration. To this end, we conduct extensive experiments to evaluate the efficiency of Q-CLIP. We first compare the total and fine-tuned parameters with several state-of-the-art VQA methods, as shown in Fig. 1. Q-CLIP requires only a minimal number of fine-tuned parameters **(0.14M)** to achieve top performance. For example, compared with MBVQA, CLIP-VQA, and Q-Align, Q-CLIP reduces the fine-tuned parameters by **666×**, **3964×**, and **58556×** respectively. Moreover, even when compared to the most efficient method, FAST-VQA, Q-CLIP still achieves a **197×** reduction in fine-tuned parameters. Additionally, Q-CLIP's total parameters remain reasonable. Compared to CLIPVQA, which is also based on VLMs, Q-CLIP has a comparable total parameter size. In contrast, Q-CLIP reduces the total parameters by **13×** compared to Q-Align, which is based on LLMs. See Appendix. B for additional efficiency experiments.

### 4.7 VISUALIZATION

To further evaluate Q-CLIP's quality perception capability, we visualize feature distributions using t-SNE (Fig. 8). From Fig. 8a and Fig. 8b, the original CLIP shows clear modality confusion, as visual and textual features overlap significantly. Additionally, the five-level prompt features are scattered and poorly separated, with some quality levels fully overlapping. This indicates that CLIP fails to effectively encode quality-relevant information. In contrast, Q-CLIP exhibits a well-structured feature space, where visual and textual modalities are distinctly separated, and quality levels form compact, aligned clusters. Such separability is key to robust cross-modal understanding in high-performing VLMs (Liang et al., 2022; Qian et al., 2023; Zhang et al., 2023a). Further analysis of video features by quality (Fig. 8c and Fig. 8d) shows that Q-CLIP clearly distinguishes high- and low-scoring videos, unlike CLIP, which presents significant overlap. These results confirm that Q-CLIP enhances the feature space through effective modality separation and structured quality encoding, enabling accurate quality perception. See Appendix. C for more visualization analysis.

## 5 CONCLUSION

In this paper, we introduce Q-CLIP, the first VQA framework built entirely upon VLMs. A Shared Cross-Modal Adapter (SCMA) is employed to optimize feature representations in both the visual and textual branches. Thanks to its minimal number of trainable parameters, SCMA significantly reduces the training cost. Additionally, a learnable five-level prompt mechanism is introduced to help the model perceive fine-grained quality variations. Furthermore, we investigate the impact of different frame sampling strategies on VQA. Experimental results demonstrate that Q-CLIP outperforms existing methods on multiple VQA datasets.

ETHICS STATEMENT

This work focuses on video quality assessment using publicly available datasets, containing no personal or sensitive information. Our method aims to improve the efficiency and accuracy of video quality evaluation for multimedia services and research. While automated video analysis could potentially be misused, we encourage responsible and ethical applications of our approach.

REPRODUCIBILITY STATEMENT

We provide all code in the supplementary material to facilitate reproducibility. All datasets used are publicly available and were used according to their respective licenses. We include detailed descriptions of model architectures, training procedures, hyperparameters, and evaluation protocols in the paper and appendix, enabling others to replicate our experiments.

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

## A    METHODS COMPARISON

To clarify the novelty of our approach, we visually compare our method with existing VQA methods that utilize VLMs in Fig. 9.

In existing VQA methods leveraging VLMs (as shown in Fig. 9(b)), VLMs serve as an auxiliary component. They provide supplementary features to enhance the performance of the backbone network. The final quality score is derived through a process of feature aggregation that combines information from both the backbone network and the VLMs.

In contrast, our method (as shown in Fig. 9(a)), which is the first VQA approach fully based on VLMs, directly employs a VLM to process the input and generate the quality score without re-

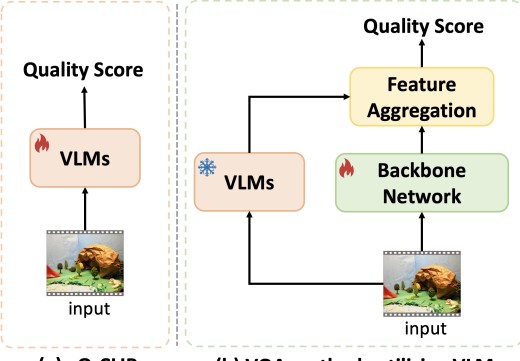

Figure 9: Methods comparison.

lying on any additional backbone network or feature aggregation with other network components. This fundamental difference highlights the originality of our work in solely harnessing the capabilities of VLMs for VQA tasks.

## B    EFFICIENCY

We conduct a comprehensive analysis of the model's runtime efficiency, as shown in Tab. 6. Specifically, we compare the FLOPs, GPU runtimes, and throughput for videos of different resolutions, where the length of the videos is 150 frames. Compared to CNN-based models (VSFA, PVQ, BVQA), MVQA-tiny reduces FLOPs by up to **6.6×**, **9.5×**, and **18.3×**, as well as reduces computation time by up to **17.1×**, **21.3×**, and **46.4×**, respectively. Furthermore, compared to methods utilizing VLMs like CLiF-VQA, MaxVQA, and CLIPVQA, our method offers significantly improved processing efficiency, being **2.7×** faster than the state-of-the-art CLIPVQA.

| Methods | 540p | | | 720p | | | 1080p | | |
|---|---|---|---|---|---|---|---|---|---|
| | FLOPs | Time | Throughput | FLOPs | Time | Throughput | FLOPs | Time | Throughput |
| VSFA | 6440 | 0.672 | 1.488 | 11426 | 1.141 | 0.876 | 25712 | 2.362 | 0.423 |
| PVQ | 9203 | 0.812 | 1.232 | 13842 | 1.334 | 0.750 | 36760 | 2.935 | 0.341 |
| BVQA | 17705 | 1.414 | 0.707 | 31533 | 3.579 | 0.279 | 70714 | 6.402 | 0.156 |
| FAST-VQA | 284 | 0.025 | 40.00 | 284 | 0.025 | 40.00 | 284 | 0.025 | 40.00 |
| DOVER | 282 | 0.031 | 32.26 | 282 | 0.031 | 32.26 | 282 | 0.031 | 32.26 |
| Q-Align | 6242 | 0.753 | 1.328 | 6242 | 0.755 | 1.325 | 6242 | 0.752 | 1.330 |
| MBVQA | 912 | 0.212 | 4.717 | 1232 | 0.249 | 4.016 | 2150 | 0.353 | 2.833 |
| CLiF-VQA | 1432 | 0.233 | 4.292 | 1432 | 0.233 | 4.292 | 1432 | 0.233 | 4.292 |
| MaxVQA | 693 | 0.161 | 6.211 | 693 | 0.161 | 6.211 | 693 | 0.161 | 6.211 |
| CLIPVQA | 3845 | 0.377 | 2.653 | 3845 | 0.376 | 2.660 | 3845 | 0.376 | 2.660 |
| **Q-CLIP** | 3870 | 0.138 | 7.246 | 3870 | 0.138 | 7.246 | 3870 | 0.138 | 7.246 |

Table 6: FLOPs, running time and throughput(average of 10 runs) on RTX 4090. FLOPs, running time and throughput are in $G$, $s$ and $videos/s$, respectively.

## C    MORE VISUALIZATIONS

We compare the attention visualizations of Q-CLIP with full fine-tuning, CLIP-Adapter, and LoRA to illustrate how Q-CLIP attends to quality-relevant regions, as shown in Fig. 10. In the visualization process, we use video frames as input to the visual encoder and provide a guiding textual prompt (such as "a video of high quality") as input to the text encoder. Each video frame is divided into multiple visual patches, and a feature vector is extracted for each patch, resulting in a patch-level visual representation of the frame. At the same time, a global semantic embedding is obtained from

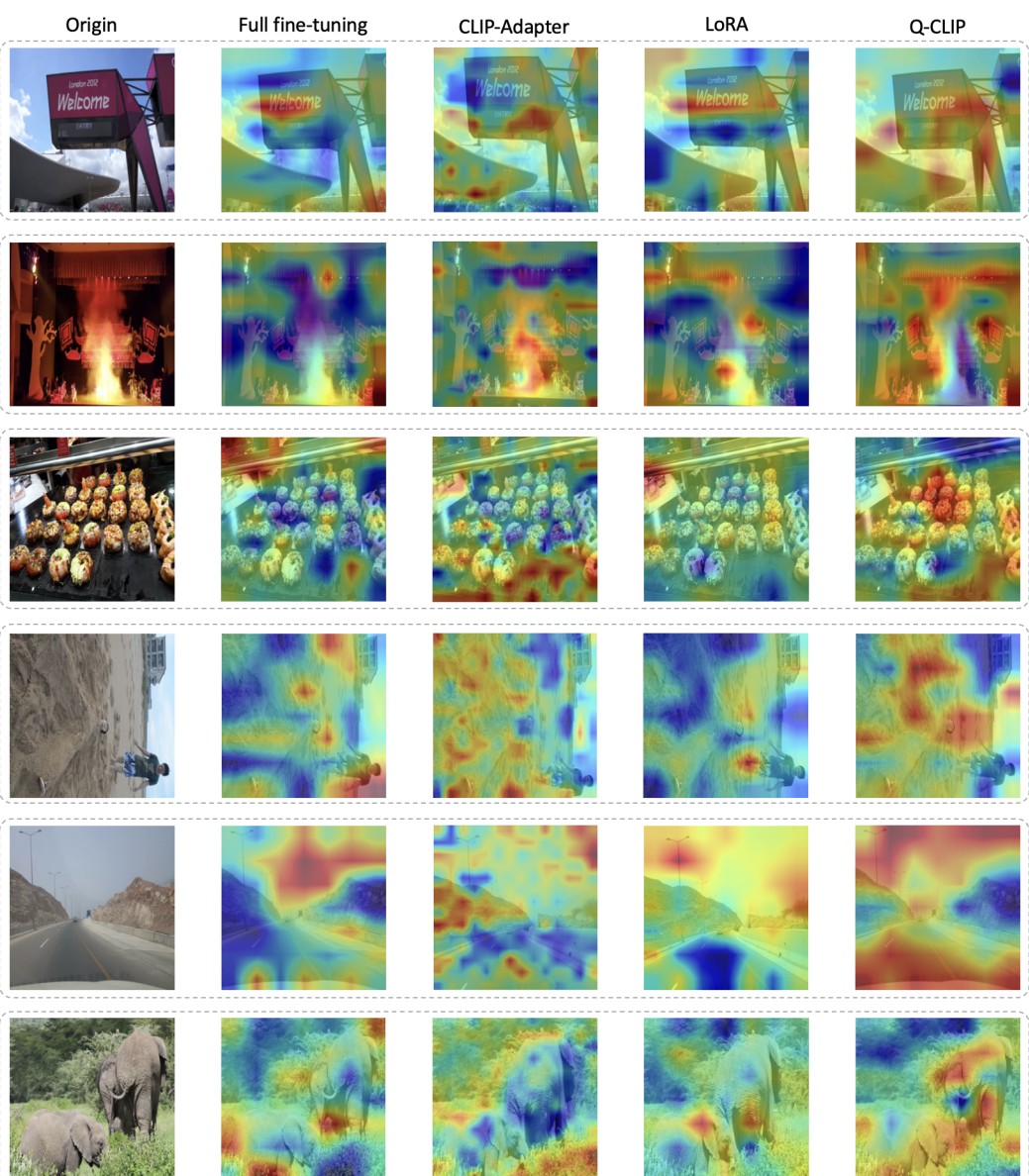

Figure 10: Comparison of attention visualizations.

the text input. We then compute the similarity between each patch feature and the text embedding, where this similarity score reflects the semantic relevance between the visual patch and the text prompt. A higher score indicates a stronger semantic association as perceived by the model. Based on these similarity scores, we construct an attention map by mapping each patch's score to a visually distinguishable color: typically, warmer colors (such as red or yellow) indicate higher attention weights, while cooler colors (such as blue or green) indicate lower ones. This results in an attention heatmap that visualizes the model's focus with respect to the given textual guidance.

From the visualization, it is evident that the attention distributions of full fine-tuning, CLIP-Adapter, and LoRA fail to align well with the attention requirements for quality perception. Full fine-tuning either scatters attention across non-critical areas or focuses on semantic regions irrelevant to quality assessment. While CLIP-Adapter and LoRA exhibit greater attention to non-semantic regions compared to full fine-tuning, they remain deficient in capturing quality-relevant regions. This is particularly notable given that such non-semantic regions inherently possess quality-relevant characteristics, yet both methods fail to adequately capture the comprehensive quality information across

the entire image. In contrast, Q-CLIP not only attends to semantic locations but also highlights multiple critical areas within video frames, resulting in a more concentrated and quality-relevant attention distribution. This suggests that, with the integration of its cross-modal adapters and other design elements, Q-CLIP achieves a more precise understanding of quality-related semantics and better captures their visual correlations. Consequently, its attention guidance for quality perception in video quality assessment is more reasonable and effective. This indicates that the model's ability to focus on critical quality-related features aligns better with the inherent characteristics of video quality evaluation, thereby enhancing both the rationality and efficacy of the assessment process.

## D MORE EXPERIMENTAL DETAILS

### D.1 ADDITIONAL MODEL DETAILS

Specifically, we add 6 layers of SCMA to the model. In the ablation study on the number of layers, the counting is done in reverse order. That is, the first SCMA layer corresponds to the last encoder layer, while the sixth SCMA layer corresponds to the sixth-to-last encoder layer. The FFN hidden dimension in both E-SCMA and P-SCMA is set to 128.

### D.2 DETAILS OF FINE-TUNING

To achieve better performance on small datasets, we adopt different fine-tuning strategies for each dataset. Specifically, for LIVE-VQC, we fine-tune only the P-SCMA module with a learning rate of 0.002. For KoNViD-1k, we train only the FFN component of E-SCMA with a learning rate of 0.001. For CVD2014 and LIVE-Qualcomm, we fine-tune only the down and up projection layers of E-SCMA, both with a learning rate of 0.001. For YouTube-UGC, we fine-tune the entire SCMA module, using separate learning rates for different components: 0.0001 for E-SCMA and 0.001 for P-SCMA. All experiments are conducted with a batch size of 12 for 20 epochs. The learning rate is scheduled using cosine annealing, with a 4-epoch warm-up phase at the beginning.

### D.3 SINGLE UNIFIED FINE-TUNING STRATEGY ALREADY ACHIEVES SOTA

In the main paper, we adopt slightly different fine-tuning hyperparameters for each small dataset (LIVE-VQC, KoNViD-1k, CVD2014, YouTube-UGC, LIVE-Qualcomm) in order to obtain a small additional performance gain. To verify that our method does not rely on such dataset-specific tuning, we further conduct experiments with a single unified fine-tuning strategy for all these datasets. Specifically, we fine-tune the entire 0.14M-parameter SCMA module using the same learning rate and training schedule on every small dataset, without any dataset-specific choices. The results in Tab. 7 show that, under this unified setting, Q-CLIP still achieves state-of-the-art performance on all small datasets and remains clearly stronger than prior VQA methods. This confirms that Q-CLIP is robust to the choice of fine-tuning strategy and that the per-dataset strategies in the main paper should be viewed as optional practical refinements rather than a requirement.

| Datasets | LIVE-VQC | | KoNViD-1K | | YouTube-UGC | | CVD2014 | | LIVE-Qualcomm | |
| --- | --- | --- | --- | --- | --- | --- | --- | --- | --- | --- |
| Methods | SROCC | PLCC | SROCC | PLCC | SROCC | PLCC | SROCC | PLCC | SROCC | PLCC |
| SCMA (Unified) | 0.879 | 0.900 | 0.913 | **0.920** | **0.911** | **0.911** | **0.898** | 0.906 | 0.844 | 0.881 |
| Q-CLIP (Per-dataset) | **0.881** | **0.901** | **0.915** | **0.920** | **0.911** | **0.911** | 0.897 | **0.907** | **0.846** | **0.884** |

Table 7: Comparison of fine-tuning strategies on small datasets.

### D.4 LOSS FUNCTION

In line with mainstream VQA methods (Wu et al., 2022; 2023a; Mi et al., 2024b; Zhao et al., 2023; Liu et al., 2024; Mi et al., 2024a), we utilize both monotonicity- and linearity-driven loss components, which are widely recognized as essential for quality prediction. The loss function used to optimize the proposed models consists of two parts: the monotonicity-induced loss and linearity-

induced loss. Given m predicted quality scores $\hat{Q} = \{\hat{q_1}, \hat{q_2}, ..., \hat{q_m}\}$ and m ground-truth subjective quality scores $Q = \{q_1, q_2, ..., q_m\}$.

Specifically, the monotonicity-induced loss predicts the monotonicity of the video quality scores by introducing additional order constraints. The monotonicity-induced loss function is defined as follows:

$$L_{mon} = \frac{1}{m^2} \sum_{i=1}^{m} \sum_{j=1}^{m} max(0, |q_i - q_j| - f(q_i, q_j) \cdot (\hat{q_i} - \hat{q_j})) \tag{12}$$

where $f(q_i, q_j) = 1$ if $q_i \geq q_j$, otherwise $f(q_i, q_j) = -1$.

In contrast, the goal of the linearity-induced loss is to compute the linear relationship between the predicted quality score and ground-truth subjective quality score. The linearity-induced loss function can be denoted as:

$$L_{lin} = (1 - \frac{\sum_{i=1}^{m}(\hat{q_i} - \hat{a})(q_i - a)}{\sqrt{\sum_{i=1}^{m}(\hat{q_i} - \hat{a})^2 \sum_{i=1}^{m}(q_i - a)^2}})/2 \tag{13}$$

where $a = \frac{1}{m}\sum_{i=1}^{m} q_i$ and $\hat{a} = \frac{1}{m}\sum_{i=1}^{m} \hat{q_i}$.

Finally, the total loss function $L$ is obtained by combining the two loss functions $L_{mon}$ and $L_{lin}$ above:

$$L = \alpha L_{mon} + \beta L_{lin} \tag{14}$$

where $\alpha$ and $\beta$ represent the weights of monotonicity-induced loss and linearity-induced loss.

## E    EXPLANATION OF SAMPLING

To provide a clearer understanding of the various sampling strategies explored in this work, we describe them in detail as follows:

**Uniform Sampling.** The video is evenly divided into 8 segments, and the middle frame of each segment is selected.

**Random Sampling.** The video is evenly divided into 8 segments, and one frame is randomly selected from each segment.

**Uniform Sampling with Random Start.** The video is evenly divided into 8 segments. A random frame is selected from the first segment, and the corresponding frame at the same relative position is selected from the remaining segments.

**Equal Intervals + Per-Segment MSE Average.** The video is evenly divided into 8 segments. From each segment, the frame whose MSE is closest to the average MSE of all frames in that segment is selected.

**Equal Intervals + Per-Segment MSE Median.** The video is evenly divided into 8 segments. From each segment, the frame with the median MSE value is selected.

**Uniform sampling over MSE-ranked frames.** All frames are ranked based on their MSE values. The ranked list is divided into 8 equal parts, and the middle frame from each part is selected.

**Definition of "Q-CLIP-Mixed" Sampling Strategy.** Q-CLIP-Mixed is a mixture of all six frame sampling strategies. It is used in both training and inference as follows:

- **Training.** For each video in each iteration, we randomly select one of the six sampling strategies with equal probability (i.e., a uniform 1/6 chance for each strategy) to extract frame subsets. In this way, the model is exposed to diverse sampling patterns and learns to be robust to how frames are selected, rather than overfitting to a single fixed strategy.

- **Inference.** We simultaneously apply all six sampling strategies to the video, then use the average of all sampled predictions as the final prediction. Therefore, Q-CLIP-Mixed can be regarded as a simple ensemble method that combines the six sampling schemes with equal weights.

| Datasets | LSVQ$_{test}$ | | LSVQ$_{1080p}$ | | KoNViD-1k | |
|---|---|---|---|---|---|---|
| P-SCMA | SROCC | PLCC | SROCC | PLCC | SROCC | PLCC |
| *w/o* | 0.886 | 0.885 | 0.815 | 0.844 | 0.877 | 0.882 |
| *w/* | **0.897** | **0.895** | **0.820** | **0.853** | **0.883** | **0.891** |

Table 8: Ablation on P-SCMA.

| Datasets | LSVQ$_{test}$ | | LSVQ$_{1080p}$ | | KoNViD-1k | |
|---|---|---|---|---|---|---|
| layers | SROCC | PLCC | SROCC | PLCC | SROCC | PLCC |
| *0* | 0.891 | 0.889 | 0.816 | 0.848 | 0.879 | 0.886 |
| *1* | **0.897** | **0.895** | **0.820** | **0.853** | **0.883** | **0.891** |
| *2* | **0.897** | **0.895** | **0.820** | 0.852 | 0.882 | 0.890 |
| *3* | 0.895 | 0.893 | 0.818 | **0.853** | 0.880 | 0.888 |
| *4* | 0.894 | 0.891 | 0.818 | 0.852 | 0.878 | 0.886 |

Table 9: Ablation on the number of FFN layers in P-SCMA.

## F   ADDITIONAL ABLATION STUDIES

### F.1   ABLATION ON P-SCMA

We validate the effectiveness of integrating P-SCMA into the projection component, as shown in Tab. 8. The results show that incorporating P-SCMA into the model significantly enhances performance. In addition, we conduct an ablation study on the number of FFN layers in P-SCMA, as shown in Tab. 9. A simple structure that only reduces and then expands the feature dimension yields suboptimal performance. Introducing a single intermediate mapping layer leads to a notable performance improvement. However, increasing the number of layers beyond one does not bring further benefits. In fact, performance begins to decline when the depth reaches three layers. This is likely because the projection is designed for a straightforward transformation to support similarity computation, and adding too many parameters may cause overfitting.

### F.2   ABLATION ON FFN DIMENSION IN SCMA

We perform an ablation study on different FFN dimensions in SCMA to assess their effect on model performance. The results are shown in Tab. 10. When the FFN dimension is set too low (below 64), the model performs poorly. As the dimension increases, the performance gradually improves, indicating that too small a dimension is insufficient for learning adequate feature representations. The best performance is achieved at a dimension of 128. However, further increasing the dimension leads to a decline in performance, suggesting that excessively high dimensions increase the number of learnable parameters and cause the model to overfit.

| Datasets | LSVQ$_{test}$ | | LSVQ$_{1080p}$ | | KoNViD-1k | |
|---|---|---|---|---|---|---|
| Dim | SROCC | PLCC | SROCC | PLCC | SROCC | PLCC |
| *16* | 0.889 | 0.888 | 0.812 | 0.843 | 0.878 | 0.887 |
| *32* | 0.894 | 0.891 | 0.815 | 0.844 | 0.880 | 0.889 |
| *64* | 0.896 | **0.895** | **0.820** | 0.850 | **0.883** | 0.890 |
| *128* | **0.897** | **0.895** | **0.820** | **0.853** | **0.883** | **0.891** |
| *256* | 0.896 | **0.895** | **0.820** | 0.852 | 0.882 | **0.891** |
| *512* | 0.893 | 0.891 | 0.818 | 0.851 | 0.879 | 0.888 |

Table 10: Ablation on FFN dimension in SCMA.

### F.3   ABLATION ON BACKBONE

To further validate that Q-CLIP's performance does not entirely depend on a specific CLIP backbone network fine-tuned on the video. We instantiate Q-CLIP with three standard image-pretrained CLIP

| Datasets | LSVQ$_{test}$ | | LSVQ$_{1080p}$ | | KoNViD-1k | | LIVE-VQC | |
|---|---|---|---|---|---|---|---|---|
| Backbone | SROCC | PLCC | SROCC | PLCC | SROCC | PLCC | SROCC | PLCC |
| CLIP-B/16 | 0.885 | 0.887 | 0.799 | 0.842 | 0.881 | 0.878 | 0.803 | 0.843 |
| CLIP-B/32 | 0.887 | 0.887 | 0.797 | 0.843 | 0.879 | 0.879 | 0.799 | 0.847 |
| CLIP-L/14 | 0.893 | 0.894 | 0.814 | 0.852 | 0.885 | 0.892 | 0.811 | 0.855 |
| **Q-CLIP** | **0.899** | **0.900** | **0.823** | **0.866** | **0.896** | **0.901** | **0.826** | **0.867** |

Table 11: Ablation on backbone.

backbones (Radford et al., 2021): CLIP-B/16, CLIP-B/32, and CLIP-L/14. For each architecture, we maintain the overall Q-CLIP framework unchanged, replacing only the text and visual encoders. We also follow the same experimental setup. We pre-train the proposed Q-CLIP on LSVQ and conduct intra-dataset testing on LSVQ$_{test}$ and LSVQ$_{1080p}$. Additionally, cross-dataset testing is performed on KoNViD-1k and LIVE-VQC. Detailed results are shown in Tab. 11. The results demonstrate that, as we progressively move from the smallest image-based backbones (CLIP-B/16 and CLIP-B/32) to slightly larger CLIP-L/14 and Q-CLIP, performance remains consistently strong. The Q-CLIP with a video-tuned backbone brings only a marginal improvement on top of this, indicating that the smaller image-pretrained CLIP backbones are already sufficient for Q-CLIP to achieve SOTA performance while offering a better efficiency–accuracy trade-off.

## F.4 LIMITED-DATA ROBUSTNESS

Evaluating performance under limited data is crucial for VQA methods. Thus, we have added extended experiments, where we randomly sub-sample the LSVQ training set to 20%, 50%, and 80%, keeping all other training settings identical to the full-data case. It is important to note that we do not employ mixed sampling to enhance model performance, but rather use the most basic uniform sampling. Detailed results are shown in Tab. 12. As can be seen, when using only 20% of the training data, Q-CLIP has achieved performance comparable to or even better than strong baselines. As the training ratio increases from 20% → 50% → 80% → 100%, the performance improves smoothly, with only marginal gains beyond 50%, indicating that Q-CLIP quickly saturates and uses labelled data efficiently. In particular, when using 80% of the training data, Q-CLIP achieves performance comparable to that obtained with 100% of the training data. These observations demonstrate that Q-CLIP is indeed robust and data-efficient under limited supervision.

| Datasets | LSVQ$_{test}$ | | LSVQ$_{1080p}$ | | KoNViD-1k | | LIVE-VQC | |
|---|---|---|---|---|---|---|---|---|
| Data Ratio | SROCC | PLCC | SROCC | PLCC | SROCC | PLCC | SROCC | PLCC |
| **Q-CLIP-20%** | 0.878 | 0.881 | 0.798 | 0.837 | 0.869 | 0.876 | 0.804 | 0.834 |
| **Q-CLIP-50%** | 0.888 | 0.890 | 0.817 | 0.850 | 0.882 | 0.889 | 0.815 | 0.844 |
| **Q-CLIP-80%** | 0.894 | **0.895** | 0.816 | 0.850 | **0.890** | **0.896** | 0.816 | **0.849** |
| **Q-CLIP-100%** | 0.897 | **0.895** | 0.820 | 0.853 | 0.883 | 0.891 | 0.803 | 0.842 |

Table 12: Ablation on limited-data.

## F.5 GENERALIZABILITY OF PROMPTS AND SAMPLING

To better isolate the core contributions of the Q-CLIP framework, we test the proposed prompt and sampling strategies on other architectures.

**(1) Learnable Five-Level Prompts on a VLMs-Utilizing Baseline (MaxVQA).**

We take MaxVQA (Wu et al., 2023c) as a representative CLIP-utilizing VQA model and re-place its original antonym-style text prompts with our learnable five-level prompts ("excellent/good/fair/poor/bad"), while keeping all other settings the same as in the MaxVQA paper. Under this setup, we re-train MaxVQA on LIVE-VQC, KoNViD-1k, and YouTube-UGC. As shown in Tab. 13. Across all three datasets, the introduction of our prompting strategy brings consistent performance improvements over the original MaxVQA. This confirms that the proposed prompt de-

sign is a generally valuable component that can benefit other CLIP-utilizing VQA models, not just Q-CLIP.

| Datasets | LIVE-VQC | | KoNViD-1k | | YouTube-UGC | |
|---|---|---|---|---|---|---|
| Methods | SROCC | PLCC | SROCC | PLCC | SROCC | PLCC |
| MaxVQA | 0.854 | 0.873 | 0.894 | 0.895 | 0.894 | 0.890 |
| **MaxVQA + Our Prompt Strategy** | **0.862** | **0.879** | **0.903** | **0.905** | **0.907** | **0.902** |

Table 13: Learnable five-level prompts on MaxVQA.

**(2) Frame Sampling on the Most Classical Baseline (FAST-VQA).**

We further apply our mixed frame sampling strategy to FAST-VQA (Wu et al., 2022), while keeping its original spatial sampling scheme unchanged. Again, using the official training settings of FAST-VQA, we evaluate on LIVE-VQC, KoNViD-1k, and YouTube-UGC. As shown in Tab. 14. The modified FAST-VQA with our sampling strategy consistently outperforms the original FAST-VQA across all three benchmarks, indicating that our sampling design is also broadly helpful for conventional video-backbone-based VQA methods.

| Datasets | LIVE-VQC | | KoNViD-1k | | YouTube-UGC | |
|---|---|---|---|---|---|---|
| Methods | SROCC | PLCC | SROCC | PLCC | SROCC | PLCC |
| FAST-VQA | 0.845 | 0.852 | 0.890 | 0.889 | 0.857 | 0.853 |
| **FAST-VQA + Our Sampling Strateg** | **0.860** | **0.862** | **0.899** | **0.900** | **0.866** | **0.868** |

Table 14: Frame sampling on FAST-VQA.

# G  THE USE OF LARGE LANGUAGE MODELS (LLMS)

We used LLMs only to correct minor grammatical errors and improve the clarity of some sentences in the manuscript. The LLM did not contribute to research ideation, experimental design, data analysis, or the interpretation of results. All substantive content and writing decisions are solely the responsibility of the authors.

