# OpenReview forum: "Q-CLIP: Unleashing the Power of Vision-Language Models for Video Quality Assessment through Unified Cross-Modal Adaptation"
_ICLR.cc/2026/Conference — Submitted to ICLR 2026_

### Official Review · Reviewer_rJe3 · 2025-10-21

**Soundness:** 3
**Presentation:** 3
**Contribution:** 2
**Rating:** 4
**Confidence:** 5

**Summary:**

This manuscript introduces Q-CLIP, a framework for VQA built on CLIP. To enhance the model's performance, the authors designed a lightweight Shared Cross-Modal Adapter (SCMA) which is the only part of the model that needs to be trained. The framework also utilizes a set of five learnable, quality-level text prompts (excellent, good, fair, poor, bad) to help the model better discern subtle differences in video quality. Additionally, the authors explore different frame-sampling strategies, finding that methods based on frame differences improve the model's generalization. Extensive experiments show that Q-CLIP achieves state-of-the-art performance on multiple VQA datasets while being computationally efficient.

**Strengths:**

1. The manuscript is generally well-written, presenting the proposed methodology and results in a clear and organized structure.

2. The experimental evaluation appears sufficient, providing empirical evidence to support the claims made regarding the model's performance.

3. The proposed model, Q-CLIP, demonstrates improvements in VQA performance while simultaneously reducing the computational costs associated with training and inference compared to existing methods

**Weaknesses:**

The claimed novelty of the VLM-based approach is undermined by an evaluation scope that neglects key VLM-specific advantages.
1. Unlike models such as Q-Align which show robustness with limited data, this capability isn't explored for Q-CLIP.
2. The potential for generating reasoned quality assessments, a feature present in models like Q-insight and VisualQuality-Q1, is not investigated.

**Questions:**

1. The claim that Q-CLIP is the "first VQA model fully based on VLMs" requires further clarification and justification. Recent methods, such as Q-Align, also appear to be constructed entirely upon VLM architectures. The authors should explicitly differentiate their framework from such existing fully VLM-based approaches to substantiate their claim of novelty.

2. The experimental validation is conducted on established VQA datasets. To further demonstrate the robustness and generalizability of Q-CLIP, it would be beneficial to evaluate its performance on more recent and diverse datasets, potentially including those with specific content types or distortions, such as gaming videos (e.g., YouTube-Gaming), fine-grained quality labels (e.g., FineVQ), and short-form content (e.g., KVQ and YouTube SFV+HDR).

3. The performance gains also attributable to the learnable prompt strategy and the frame-difference sampling approach. However, these components could arguably be integrated into prior learning-based or even traditional VQA architectures. To isolate the contribution of the core Q-CLIP framework, ablation studies applying the proposed prompting and sampling techniques to other existing VQA models would strengthen the paper's claims regarding the specific benefits of the proposed architecture.

---

> ### Author Response · Authors · 2025-11-19
> **Question 1: Limited-Data Robustness**
>
> ## We sincerely thank the reviewer for the detailed and thoughtful comments. Below, we would like to respond to each concern in turn.
>
> ## **Question 1: Limited-Data Robustness**
>
> ## **Answer：**
> We fully agree that evaluating performance under limited data is crucial for VQA methods. Thus, we have added extended experiments, **where we randomly sub-sample the LSVQ training set to 20%, 50%, and 80%**, keeping all other training settings identical to the full-data case. It is important to note that we do not employ mixed sampling to enhance model performance, but rather use the most basic uniform sampling. The resulting models, denoted Q-CLIP-20/50/80/100%, are compared with Q-Align, PTM-VQA, CLiF-VQA, and CLIPVQA. Detailed results are shown in the table below:
>
> | Methods         | LSVQ$_{test}$ | LSVQ$_{1080p}$ |  KoNViD-1K  |   LIVE-VQC  |
> |-----------------|:-------------:|:--------------:|:-----------:|:-----------:|
> | Q-Align         |  0.883/0.882  |   0.797/0.830  | 0.865/0.877 |     -/-     |
> | PTM-VQA         |  0.855/0.864  |   0.736/0.782  | 0.824/0.830 | 0.785/0.737 |
> | CLiF-VQA        |  0.886/0.887  |   0.790/0.832  | 0.877/0.874 | 0.834/0.855 |
> | CLIPVQA         |  0.881/0.883  |   0.782/0.827  | 0.864/0.887 | 0.781/0.871 |
> | **Q-CILP-20%**  |  0.878/0.881  |   0.798/0.837  | 0.869/0.876 | 0.804/0.834 |
> | **Q-CILP-50%**  |  0.888/0.890  |   0.817/0.850  | 0.882/0.889 | 0.815/0.844 |
> | **Q-CILP-80%**  |  0.894/0.895  |   0.816/0.850  | 0.890/0.896 | 0.816/0.849 |
> | **Q-CILP-100%** |  0.897/0.895  |   0.820/0.853  | 0.883/0.891 | 0.803/0.842 |
>
>
> As can be seen, **when using only 20% of the training data, Q-CLIP has achieved performance comparable to or even better than strong baselines.** **As the training ratio increases from 20% → 50% → 80% → 100%, the performance improves smoothly, with only marginal gains beyond 50%**, indicating that Q-CLIP quickly saturates and uses labelled data efficiently. These observations demonstrate that Q-CLIP is indeed robust and data-efficient under limited supervision.

---

> ### Author Response · Authors · 2025-11-19
> **Question 2: Generating Reasoned Quality Assessment**
>
> ## **Question 2: Generating Reasoned Quality Assessment**
>
> ## **Answer：**
> We fully agree that generating textual rationales (as in Q-Insight and VisualQuality-Q1) is a valuable research direction. However, we would also like to clarify that:
>
> ### **(1) Fundamental Architectural Difference (Discriminative vs. Generative).**
>
> - Q-Insight and VisualQuality-R1 are built upon Generative **Multi-modal Large Language Models (MLLMs) (e.g., Qwen-2.5-VL-7B) and trained with GRPO-style reinforcement learning.** These architectures inherently **include heavy decoders designed for token-by-token text generation**, making "reasoning" (generating text explanations) a native feature.
>
> - Q-CLIP, in contrast, is built upon CLIP, which is a **Contrastive Vision-Language architecture**. **It is designed to align visual and textual embeddings for discriminative tasks, not to generate text sequences.** Enabling textual reasoning in a CLIP-based framework would require attaching a heavy language decoder, which would fundamentally transform the model into an MLLM. **This falls outside the scope of our research, which aims to unleash the power of existing contrastive VLMs efficiently.**
>
>
> ### **(2) Visual Reasoning and Interpretability in Q-CLIP.**
> While Q-CLIP does not output textual reasoning chains, it provides interpretability through alternative mechanisms tailored for discriminative models. **Instead of outputting only a scalar MOS, Q-CLIP predicts a similarity distribution over five textual quality anchors** ("excellent/good/fair/poor/bad"). This plays a similar role to the textual rationales in Q-Insight/VisualQuality-R1, but in a discrete and extremely lightweight form.
>
>
> ### **(3) Compatible with Reasoning-Style Extensions.**
> Even though Q-CLIP does not output natural language explanations, it already produces: **(1) quality-aware cross-modal embeddings via SCMA + quality prompts, and (2) quality-focused attention maps showing that the model attends to quality-relevant regions beyond purely semantic objects.** These outputs can naturally serve as inputs for inference modules (such as large language models) in future work, aligning closely with the concepts of Q-Insight and VisualQuality-R1. However, Q-CLIP plays the role of a specialized and efficient video quality encoder in this context. Specifically, the following can be done: Input video-level embeddings and attention-pooled frame tokens extracted via Q-CLIP into a large language model (LLM), and train this model using SFT or GRPO to generate textual explanations, similar to Q-Insight/VisualQuality-R1.
>
>
> We appreciate the reviewer for highlighting this as a valuable and promising research direction. In the revised manuscript, we will more clearly articulate Q-CLIP's potential for supporting reasoned quality assessment.

---

> ### Author Response · Authors · 2025-11-19
> **Question 3: Clarification on "First VQA Model Fully Based on VLMs"**
>
> ## **Question 3: Clarification on "First VQA Model Fully Based on VLMs"**
>
> ## **Answer：**
> **Our intention with the phrase "fully based on VLMs" was to distinguish CLIP-type contrastive VLMs (non-generative encoders) from Large Language Models (LLMs) with generative language capabilities.** Of course, we acknowledge that Q-Align incorporates both visual encoders and text encoders, and it seems reasonable to classify it as a VLM from this perspective. However, we believe that classifying models solely based on the type of data they process is not rigorous. A rigorous and accurate classification should be based on the model's architecture and underlying principles. Therefore, we categorize methods based on their core architectural design, and **Q-Align and Q-CLIP exhibit fundamental differences in their core architectures:**
>
>
> - **Q-Align explicitly positions itself as “teaching LMMs for visual scoring via discrete text-defined levels” and operates under the original LMMs structure**, where visual tokens are fed into a multimodal large language model that performs the core reasoning and scoring. **In this sense, we regard Q-Align as an LLMs-based quality assessment method.**
>
>
> - **In contrast, Q-CLIP is built strictly on a CLIP-style, contrastively trained VLMs**: we use only the image encoder and text encoder of a CLIP-like model, keep them entirely frozen, and introduce a single Shared Cross-Modal Adapter (SCMA, 0.14M parameters) as the only trainable module that jointly adapts both modalities for VQA.
>
> **Therefore, we classify Q-Align under the “LLMs-based” category, while Q-CLIP falls under VLMs.** We agree that our original phrasing could be interpreted too broadly and thus appear to conflict with Q-Align. **In the revised version, we will clarify this to eliminate any ambiguity.**

---

> > ### Author Response · Authors · 2025-11-27
> >
> > ### **Additional evidence for our use of “VLMs”.**
> >
> > To further clarify the definition of VLMs, we have compiled a substantial set of recent works [1-18] as supporting evidence. **In these papers, the term vision-language models (VLMs) is consistently used to refer to CLIP-type, contrastively trained dual-encoder architectures. In other words, our paper follows an established convention in the community rather than introducing a non-standard definition: “VLMs” typically denotes CLIP-type contrastive models.**
> >
> > Although Q-Align involves both visual and textual modalities, it fundamentally relies on the generative capability of a Large Language Model (LLMs) to produce discrete rating levels. The Q-Align paper explicitly positions itself as a method based on LMMs, rather than a CLIP-type VLMs approach.
> >
> > [1] Wasim, Syed Talal, et al. "Vita-clip: Video and text adaptive clip via multimodal prompting." Proceedings of the IEEE/CVF Conference on Computer Vision and Pattern Recognition (CVPR). 2023.
> >
> > [2] Lu, Yuning, et al. "Prompt distribution learning." Proceedings of the IEEE/CVF Conference on Computer Vision and Pattern Recognition (CVPR). 2022.
> >
> > [3] Menon, Sachit, and Carl Vondrick. "Visual Classification via Description from Large Language Models." The Eleventh International Conference on Learning Representations (ICLR).
> >
> > [4] Shu, Manli, et al. "Test-time prompt tuning for zero-shot generalization in vision-language models." Advances in Neural Information Processing Systems (NIPS) 35 (2022): 14274-14289.
> >
> > [5] Wu, Wenhao, Zhun Sun, and Wanli Ouyang. "Revisiting classifier: Transferring vision-language models for video recognition." Proceedings of the AAAI Conference on Artificial Intelligence (AAAI). Vol. 37. No. 3. 2023.
> >
> > [6] Rasheed, Hanoona, et al. "Fine-tuned clip models are efficient video learners." Proceedings of the IEEE/CVF Conference on Computer Vision and Pattern Recognition (CVPR). 2023.
> >
> > [7] Khattak, Muhammad Uzair, et al. "Maple: Multi-modal prompt learning." Proceedings of the IEEE/CVF Conference on Computer Vision and Pattern Recognition (CVPR). 2023.
> >
> > [8] Lee, Dongjun, et al. "Read-only prompt optimization for vision-language few-shot learning." Proceedings of the IEEE/CVF International Conference on Computer Vision (ICCV). 2023.
> >
> > [9] Yao, Hantao, Rui Zhang, and Changsheng Xu. "Visual-language prompt tuning with knowledge-guided context optimization." Proceedings of the IEEE/CVF Conference on Computer Vision and Pattern Recognition (CVPR). 2023.
> >
> > [10] Singha, Mainak, et al. "Ad-clip: Adapting domains in prompt space using clip." Proceedings of the IEEE/CVF International Conference on Computer Vision (ICCV). 2023.
> >
> > [11] Udandarao, Vishaal, Ankush Gupta, and Samuel Albanie. "Sus-x: Training-free name-only transfer of vision-language models." Proceedings of the IEEE/CVF International Conference on Computer Vision (ICCV). 2023.
> >
> > [12] Chen, Guangyi, et al. "PLOT: Prompt Learning with Optimal Transport for Vision-Language Models." The Eleventh International Conference on Learning Representations (ICLR).
> >
> > [13] Ren, Shuhuai, et al. "Prompt pre-training with twenty-thousand classes for open-vocabulary visual recognition." Advances in Neural Information Processing Systems (NIPS) 36 (2023): 12569-12588.
> >
> > [14] Gao, Peng, et al. "Clip-adapter: Better vision-language models with feature adapters." International Journal of Computer Vision (IJCV) 132.2 (2024): 581-595.
> >
> > [15] Xu, Chen, et al. "Progressive visual prompt learning with contrastive feature re-formation." International Journal of Computer Vision (IJCV) 133.2 (2025): 511-526.
> >
> > [16] Abdul Samadh, Jameel, et al. "Align your prompts: Test-time prompting with distribution alignment for zero-shot generalization." Advances in Neural Information Processing Systems (NIPS) 36 (2023): 80396-80413.
> >
> > [17] Zhu, Yuhan, et al. "Awt: Transferring vision-language models via augmentation, weighting, and transportation." Advances in Neural Information Processing Systems (NIPS) 37 (2024): 25561-25591.
> >
> > [18] Khattak, Muhammad Uzair, et al. "Learning to prompt with text only supervision for vision-language models." Proceedings of the AAAI Conference on Artificial Intelligence (AAAI). Vol. 39. No. 4. 2025.

---

> ### Author Response · Authors · 2025-11-19
> **Question 4: Experiments on Additional Datasets**
>
> ## **Question 4: Experiments on Additional Datasets**
> ## **Answer：**
> We agree that evaluating Q-CLIP on more datasets would better showcase its robustness; however, because the datasets suggested by the reviewer are relatively new and not yet widely adopted, we encountered practical difficulties in obtaining all of them. Due to dataset availability, we were currently only able to obtain KVQ and YouTube SFV+HDR. The official download link for YouTube-Gaming appears to be no longer accessible, and FineVQ requires an access application, which we have already submitted but has not yet been approved. **Therefore, we report the additional experiments available on KVQ and YouTube SFV+HDR, and we will provide results on the other two datasets in this revision if possible.**
>
>
> | Methods    |     KVQ     | YouTube SFV+HDR |
> |------------|:-----------:|:---------------:|
> | FASTVQA    | 0.832/0.834 |   0.752/0.797   |
> | DOVER      | 0.833/0.837 |   0.702/0.781   |
> | KSVQE      | 0.867/0.869 |       -/-       |
> | **Q-CLIP** | **0.877/0.878** |   **0.804/0.825**   |
>
> The results demonstrate that **CLIP continues to deliver outstanding performance on both datasets, further highlighting its robustness across diverse content types and distortion patterns.**

---

> ### Author Response · Authors · 2025-11-19
> **Question 5: Generalizability of Prompts and Sampling**
>
> ## **Question 5: Generalizability of Prompts and Sampling**
> ## **Answer：**
> We appreciate the reviewer’s insightful suggestion to test the proposed prompting and sampling strategies on other VQA architectures in order better to isolate the contribution of the core Q-CLIP framework, and we have conducted additional experiments accordingly.
>
> ### **(1) Learnable Five-Level Prompts on a VLMs-Utilizing Baseline (MaXVQA).**
> We take MaXVQA [1] as a representative CLIP-utilizing VQA model and replace its original antonym-style text prompts with our learnable five-level prompts ("excellent/good/fair/poor/bad"), while keeping all other settings the same as in the MaXVQA paper. Under this setup, we re-train MaXVQA on LIVE-VQC, KoNViD-1k, and YouTube-UGC. **Across all three datasets, the introduction of our prompting strategy brings consistent performance improvements over the original MaXVQA.** This confirms that the proposed prompt design is a generally valuable component that can benefit other CLIP-utilizing VQA models, not just Q-CLIP.
>
> | Methods                     |   LIVE-VQC  |  KoNViD-1K  | YouTube-UGC |
> |-----------------------------|:-----------:|:-----------:|:-----------:|
> | MaXVQA                      | 0.854/0.873 | 0.894/0.895 | 0.894/0.890 |
> | **MaXVQA + Our Prompt Strategy** | **0.862/0.879** | **0.903/0.905** | **0.907/0.902** |
>
>
> ### **(2) Frame Sampling on the Most Classical Baseline (FAST-VQA).**
> We further apply our mixed frame sampling strategy to FAST-VQA [2], while keeping its original spatial sampling scheme unchanged. Again, using the official training settings of FAST-VQA, we evaluate on LIVE-VQC, KoNViD-1k, and YouTube-UGC. **The modified FAST-VQA with our sampling strategy consistently outperforms the original FAST-VQA across all three benchmarks**, indicating that our sampling design is also broadly helpful for conventional video-backbone-based VQA methods.
>
> | Methods                     |   LIVE-VQC  |  KoNViD-1K  | YouTube-UGC |
> |-----------------------------|:-----------:|:-----------:|:-----------:|
> | FAST-VQA                      | 0.845/0.852 | 0.890/0.889 | 0.857/0.853 |
> | **FAST-VQA + Our Sampling Strategy** | **0.860/0.862** | **0.899/0.900** | **0.866/0.868** |
>
> ### References:
> [1] Wu, Haoning, et al. "Towards explainable in-the-wild video quality assessment: a database and a language-prompted approach." Proceedings of the 31st ACM International Conference on Multimedia (ACMMM). 2023.
>
> [2] Wu, Haoning, et al. "Fast-vqa: Efficient end-to-end video quality assessment with fragment sampling." European Conference on Computer Vision (ECCV). Cham: Springer Nature Switzerland, 2022.

---

> ### Comment · Reviewer_rJe3 · 2025-11-27
> **Response**
>
> I appreciate the authors' thorough response to the prior concerns.
>
> I am particularly impressed by the results of the added experiments, specifically those involving video processing. Given the computational expense associated with processing video data in LLMs and VLMs, the demonstrated performance is a significant strength of this work.
>
> While the new experiments are highly valuable, I still have a minor concern regarding the novelty and clear distinction between the core concepts of LLMs and VLMs, as well as generative and discriminative models, within the current manuscript's presentation.
>
> Based on the new experiments, I have raised my rating.
>
> The authors are required to integrate all related experiments and analysis discussed in their response into the revised manuscript to ensure the full scope of the work is available for final evaluation. This integration should also include an effort to clarify the conceptual boundaries of the modeling approaches used.

---

> > ### Author Response · Authors · 2025-11-29
> >
> > Thank you very much for your thoughtful follow-up and for raising your rating. We are glad that you found the new experiments, especially the video processing results, helpful and convincing.
> >
> > We fully acknowledge your remaining concern about the clarity of the differences between LLMs and VLMs. In the revised manuscript, we will clearly define these terms, make the distinction between these model families more explicit, and sharpen the positioning and novelty of our method under these definitions.
> >
> > As requested, we will also integrate all additional experiments and analyses from our response into the revised version so that the full scope of the work is visible for the final evaluation.
> >
> > We sincerely thank you again for your constructive comments and support.

---

### Official Review · Reviewer_HNXL · 2025-10-27

**Soundness:** 3
**Presentation:** 3
**Contribution:** 3
**Rating:** 6
**Confidence:** 5

**Summary:**

This paper proposes Q-CLIP, a pioneering Video Quality Assessment (VQA) framework built entirely on Vision-Language Models (VLMs). It addresses two long-standing challenges in mainstream VQA methods: first, classification-based pretraining fails to capture multi-dimensional video quality factors (semantics, distortion, motion, aesthetics); second, large-scale pretraining requires excessive computational costs. Q-CLIP introduces key innovations to solve these issues. It includes a lightweight Shared Cross-Modal Adapter (SCMA) with only 0.14M trainable parameters and a set of learnable five-level quality prompts. These designs enable Q-CLIP to outperform state-of-the-art baselines across all tested datasets. Additionally, the paper explores the impact of frame-difference-based sampling on VQA performance. This pioneering exploration provides valuable insights for future research. Rigorous ablation studies, efficiency analyses, and visualizations further validate the rationality and superiority of Q-CLIP’s design.

**Strengths:**

(1) Clear writing and well-defined motivation
The paper follows a logical structure. Methods are clearly explained using figures and equations, and details of the model and training processes are thoroughly described, this makes reproducibility straightforward. The motivation of the work is also well-defined: it aims to solve two key problems in existing studies. One is that semantic knowledge from classification pretraining cannot capture multi-dimensional quality factors. The other is that large-scale pretraining leads to high computational costs.

(2) Novel technical design
Most previous VQA methods that use VLMs treat VLMs as auxiliary feature extractors. In contrast, Q-CLIP constructs the first VQA architecture fully based on VLMs. This design eliminates reliance on external backbone networks and fully leverages the cross-modal representation capabilities of VLMs. It is a critical breakthrough that redefines how VLMs can be applied to VQA tasks.

(3) Comprehensive and convincing experiments
The experimental design is comprehensive and reliable. Experiments are conducted on a wide range of datasets, and results are compared with over 15 baselines (including knowledge-driven, data-driven, and VLMs-based models). This clearly demonstrates the effectiveness of Q-CLIP. Furthermore, ablation studies further confirm the rationality of Q-CLIP’s design. In addition, thorough efficiency experiments prove that Q-CLIP is highly efficient.

(4) Excellent parameter optimization
Q-CLIP’s clever and novel shared cross-modal adapter design greatly reduces the number of trainable parameters. Only 0.14M parameters need training, which highlights the high efficiency of Q-CLIP.
(5) Pioneering exploration of frame sampling strategies
Most previous VQA methods rely on random or uniform sampling. This work is the first to explore the impact of frame-difference-based sampling on VQA. It provides actionable insights for future research in this area.

**Weaknesses:**

(1) In the comparison of fine-tuning methods, some approaches are missing. Examples include COOP and VPT. Although these methods may not perform well in VQA tasks, adding comparisons with them would make the experimental results more comprehensive.

(2) Although efficiency experiments demonstrate that Q-CLIP outperforms most baseline methods, I believe there is still room for optimization in its parameter scale. Attempting to use smaller backbone networks may help achieve a better balance between performance and efficiency.

(3) The paper contains a few minor writing errors. For instance, there is an extra "s" in line 317. In line 467, the references to "Fig. 6(a)" and "Fig. 6(b)" are inconsistent with the actual figure numbering (should correspond to Fig. 8 based on the context). These errors should be corrected to improve the paper’s accuracy.

**Questions:**

Please see the weaknesses.

---

> ### Author Response · Authors · 2025-11-17
> **Question 1: Additional Comparisons with CoOp and VPT**
>
> ## We sincerely thank the reviewer for the positive and constructive feedback. We address the three concerns as follows:
>
> ## **Question 1: Additional Comparisons with CoOp and VPT**
>
> ## **Answer：**
> In response to the reviewer’s suggestion, we have added experiments comparing Q-CLIP with CoOp and VPT under the same CLIP backbone, as shown in the table below. As can be seen, **CoOp performs poorly (0.763/0.764)**, which is expected since it only optimizes prompts while keeping all backbone parameters frozen, and thus cannot sufficiently adapt the representations to fine-grained video quality. **VPT achieves acceptable results (0.823/0.820) by introducing visual prompt tokens, but it is still clearly behind CLIP-Adapter/LoRA and our method.**
>
> By contrast, Q-CLIP achieves 0.897/0.895, outperforming CoOp, VPT, full fine-tuning, CLIP-Adapter, and LoRA on the same backbone. **These results further validate the effectiveness of the method we proposed.**
>
> | Methods          |    SROCC/PLCC   |
> |------------------|:---------------:|
> | CoOp             |   0.763/0.764   |
> | VPT              |   0.823/0.820   |
> | Full fine-tuning |   0.816/0.811   |
> | CLIP-Adapter     |   0.881/0.884   |
> | LoRA             |   0.883/0.883   |
> | **Q-CLIP**       | **0.897/0.895** |

---

> ### Author Response · Authors · 2025-11-17
> **Question 2: Using Smaller Backbones for Better Efficiency–Performance Trade-Off**
>
> ## **Question 2: Using Smaller Backbones for Better Efficiency–Performance Trade-Off**
> ## **Answer：**
> We sincerely agree that exploring smaller backbones is essential. To this end, we have instantiated Q-CLIP with three standard image-pretrained CLIP backbones [1]: CLIP-B/16, CLIP-B/32, and CLIP-L/14. For each architecture, we maintain the overall Q-CLIP framework unchanged, replacing only the text and visual encoders. We also follow the same experimental setup. We pre-train the proposed Q-CLIP on LSVQ and conduct intra-dataset testing on $LSVQ_{test}$ and $LSVQ_{1080p}$. Additionally, cross-dataset testing was performed on KoNViD-1k and LIVE-VQC. Detailed results are shown in the table below:
>
> | Methods       | LSVQ$_{test}$ | LSVQ$_{1080p}$ |  KoNViD-1K  |   LIVE-VQC  |
> |---------------|:-------------:|:--------------:|:-----------:|:-----------:|
> | Q-Align       |  0.883/0.882  |   0.797/0.830  | 0.865/0.877 |     -/-     |
> | PTM-VQA       |  0.855/0.864  |   0.736/0.782  | 0.824/0.830 | 0.785/0.737 |
> | CLiF-VQA      |  0.886/0.887  |   0.790/0.832  | 0.877/0.874 | 0.834/0.855 |
> | CLIPVQA       |  0.881/0.883  |   0.782/0.827  | 0.864/0.887 | 0.781/0.871 |
> | **CLIP-B/16** |  0.885/**0.887**  |   **0.799**/**0.842**  | **0.881**/0.878 | 0.803/0.843 |
> | **CLIP-B/32** |  **0.887**/**0.887**  |   **0.797**/**0.843** | **0.879**/0.879 | 0.799/0.847 |
> | **CLIP-L/14** |  **0.893**/**0.894**  |   **0.814**/**0.852**  | **0.885**/**0.892** | 0.811/0.855 |
> | **Q-CLIP**    |  **0.899**/**0.900**  |   **0.823**/**0.866** | **0.896**/**0.901** | 0.826/0.867 |
>
> The results demonstrate that, as we progressively move from the smallest image-based backbones (CLIP-B/16 and CLIP-B/32) to slightly larger CLIP-L/14 and Q-CLIP, performance remains consistently strong. **The Q-CLIP with a video-tuned backbone brings only a marginal improvement on top of this, indicating that the smaller image-pretrained CLIP backbones are already sufficient for Q-CLIP to achieve SOTA performance while offering a better efficiency–accuracy trade-off.**
>
> ### References:
> [1] Radford, Alec, et al. "Learning transferable visual models from natural language supervision." International Conference on Machine Learning. PMLR, 2021.

---

> ### Author Response · Authors · 2025-11-17
> **Question 3: Minor Writing Issues**
>
> ## **Question 3: Minor Writing Issues**
> ## **Answer：**
> We truly thank the reviewer for the careful reading and for pointing out these minor writing issues. **We will correct these typographical errors in the revised version and perform an additional proofreading pass to improve the overall clarity and accuracy of the manuscript.**

---

### Official Review · Reviewer_AvtS · 2025-10-29

**Soundness:** 2
**Presentation:** 3
**Contribution:** 2
**Rating:** 4
**Confidence:** 4

**Summary:**

This work proposes Q-CLIP, a VLMs-based framework for VQA. Q-CLIP enhances both visual and textual representations through an SCMA module. This work also introduces a set of five learnable quality-level prompts to guide the VLMs in perceiving subtle quality variations.

**Strengths:**

1. This work achieves better performance with smaller parameters on VQA datasets.
2. This submission is well-written and easy to follow.

**Weaknesses:**

1. The novelty of this work is somewhat weak. This work is somewhat like combining the techniques proposed in CLIP-IQA (text-vision similarity), Q-Align (five rating levels), together with common learnable prompts. Besides, the difference-based sampling method is also simple and intuitive, which is more like a baseline method.
2. Compared with the original CLIP, the main revision is the proposed SCMA module. However, the reason why this module works is not well analyzed or supported.
3. In Table 1, Q-Align should also be under the type of VLMs, instead of LLMs, because Q-Align supports both vision and text inputs and could well understand vision modality.
4. Another concern is about the encoding of temporal information. This work extracts features for different frames separately and then averages these features to obtain the video feature. However, this operation ignores the temporal information. For example, for one video, all frames are good-quality images, but there are quite large inter-frame inconsistencies; the proposed model will mistakenly assess this video as a high-quality video.
5. This work has conducted an ablation study to show that the mean pooling is better than the transformer or mamba architecture. However, after encoding all frames into features, the temporal information (like inter-frame inconsistencies) has been almost lost. Therefore, the fusion should be performed in a less shallow position, where some pixel-level information is well kept.

**Questions:**

I would like the authors to (1) clarify the novelty of this work, (2) support the proposed SCMA module, and (3) show how to assess the temporal quality of videos.

---

> ### Author Response · Authors · 2025-11-19
> **Question 1: Novelty Clarification**
>
> ## We sincerely thank the reviewer for the careful reading and constructive comments. Meanwhile, we would like to address the concerns raised, as detailed below:
>
> ## **Question 1: Novelty Clarification**
>
> ## **Answer：**
> While we appreciate this perspective, **we believe that the characterization of Q-CLIP as “somewhat like combining CLIP-IQA (text–vision similarity), Q-Align (five rating levels), together with common learnable prompts” does not fully capture our method**. Below, we clarify the distinctions and our specific contributions.
>
> ### **(1) Relation to CLIP-IQA.**
> CLIP-IQA mainly explores how far a frozen CLIP can go for **Image Quality Assessment (IQA)** by directly computing text–image similarity, with antonym prompts (e.g., "good quality" vs. "bad quality") but without any fine-tuning of the CLIP backbone. **However, we enhance the performance of CLIP-type VLMs in VQA by introducing a Shared Cross-Modal Adapter (SCMA) and learnable five-level prompts. Therefore, Q-CLIP fundamentally differs from CLIP-IQA.**
>
>
> ### **(2) Relation to Q-Align.**
> Q-Align is built upon Large Language Models (LLMs). It discretizes labels into five rating levels and leverages the generative capabilities of LLMs to directly generate descriptions of quality levels. **In contrast, Q-CLIP is built on contrastively trained, CLIP-type VLMs (non-generative). We keep both the image and text encoders frozen and adapt them only through SCMA. The five learnable prompts act as textual anchors in the embedding space, and the final quality score is obtained by automatically fitting the MOS through similarity-based regression over these anchors, not by generative prediction.**
>
> Due to the fundamentally different model architectures, where Q-Align relies on LLMs while Q-CLIP relies on CLIP-type VLMs, we classify Q-Align as an LLMs-based method in this paper. This classification also addresses the reviewer's third question: “Q-Align should also be categorized under VLMs.”
>
> ### **(3) Our Main Novelty.**
>
> - **SCMA as A New Fine-Tuning Strategy for CLIP-type VLMs in VQA Tasks.**
> SCMA completely departs from the commonly used CLIP-Adapter/LoRA-style strategies: it is a single, shared cross-modal adapter applied to both the visual and textual branches, with microscopic trainable parameters, yet it significantly improves VQA performance. Our experiments show that this SCMA-based adaptation outperforms widely used approaches such as CLIP-Adapter and LoRA in the QA setting, while using fewer parameters. This is, to our knowledge, a new way of adapting CLIP-type VLMs.
>
> - **Learnable Five-Level Prompts Integrated Into Contrastive VLMs.**
> While the idea of discrete quality levels exists in Q-Align, our formulation is essentially different: Q-Align uses an LLM to generate one of the discrete ratings, whereas we leverage the similarity structure of CLIP-type VLMs and learn five textual anchors that, together with SCMA, automatically fit continuous MOS via similarity-based regression. In other words, we do not merely adopt “five levels” from Q-Align; we design a way to embed these levels into contrastive VLMs and show that this leads to strong and efficient VQA performance.
>
> ### **(4) Frame-Difference-Based Sampling.**
> We acknowledge that the proposed frame sampling method is intuitive and straightforward, and we do not position it as the core innovation of this paper. Our motivation for studying this method stems from the fact that, in current VQA research, studies on spatial sampling of frames are already relatively abundant, while temporal sampling (i.e., how to select frames) has received little attention. This research aims to fill this gap in the field. Therefore, we regard frame sampling as a contribution rather than a primary innovation: its purpose is to provide concrete empirical evidence and a concise benchmark for subsequent research to build upon, thereby drawing greater academic attention to the significance of frame sampling in VQA.

---

> ### Author Response · Authors · 2025-11-19
> **Question 2: Supporting and Analyzing SCMA**
>
> ## **Question 2: Supporting and Analyzing SCMA**
>
> ## **Answer：**
>
> We will conduct a detailed discussion on the design concept and working principle of SCMA from the following three aspects.
>
> ### **(1) Design Motivation.**
> **Existing studies [1-3] have shown that the performance of VLMs on downstream tasks is often limited by the modality gap between visual and textual representations**: CLIP-type models are pretrained to align semantics, but downstream tasks (especially quality assessment) depend on additional factors such as distortion severity. Within the pre-trained representation space of VLMs, two videos with identical semantics but very different perceptual quality can still be close, and quality-critical directions (e.g., noise level, blockiness, flicker) may be weakly expressed. **Furthermore, conventional fine-tuning often leads to divergent optimization of visual and textual features, thereby undermining the prior information embedded in CLIP's original feature space.**
>
> SCMA is introduced to explicitly address these issues. **It is a lightweight cross-modal adapter implemented as an MLP whose weights are shared between the visual and textual branches.** This design forces both modalities to pass through exactly the same linear transformation and thus constrains the model to learn a common quality-aware correction of VLMs features instead of two separate, modality-specific mappings. As a result, the transformed visual and textual features remain in a well-aligned metric space that still respects VLMs’ semantic structure, while becoming more sensitive to quality-related factors. **In addition, SCMA employs both inter-layer sharing and cross-modal sharing of parameters, which greatly reduces the number of tunable parameters, preserves more of VLMs’ original knowledge, and improves generalization across diverse VQA datasets and distortion patterns.**
>
> ### References:
> [1] Liang, Victor Weixin, et al. "Mind the gap: Understanding the modality gap in multi-modal contrastive representation learning." Advances in Neural Information Processing Systems (NIPS) 35 (2022): 17612-17625.
>
> [2] Qian, Qi, Yuanhong Xu, and Juhua Hu. "Intra-modal proxy learning for zero-shot visual categorization with clip." Advances in Neural Information Processing Systems (NIPS) 36 (2023): 25461-25474.
>
> [3] Zhang, Yuhui, et al. "Diagnosing and Rectifying Vision Models using Language." The Eleventh International Conference on Learning Representations (ICML).
>
>
> ### **(2) Theoretical View.**
> From the representation learning perspective, we can view SCMA as a shared residual transformation of the joint space in VLMs.
>
> Let $f_v(x)$ and $f_t(y)$ denote the frozen visual and textual embeddings from the CLIP backbone. SCMA applies the same residual adapter $A(\cdot)$ to both modalities:
>
> $z_v = f_v(x) + A\big(f_v(x)\big), z_t = f_t(y) + A\big(f_t(y)\big)$
>
>
> The quality-related similarity used by Q-CLIP is then $\langle z_v, z_t \rangle$. If we locally linearize the adapter (a standard first-order approximation), we may view $A(f) \approx J f + b$ around the current feature region, leading to a shared linear transform $T = I + J$ and
> $\langle z_v, z_t \rangle
> \approx \langle T f_v, T f_t \rangle
> = f_v^\top (T^\top T) f_t$
>
> Thus, in this local view, SCMA effectively applies a shared linear transform $T$ to the frozen CLIP embeddings, inducing a Mahalanobis-style metric $M = T^\top T$ in the joint space. Training SCMA with MOS supervision can therefore be interpreted as learning a quality-aware metric on top of CLIP, rather than re-learning the entire encoder. Because the same adapter is shared between visual and textual branches, both modalities are forced to live in the same transformed coordinate system, which explicitly reduces the visual–textual gap for quality. The five quality-level prompts (excellent/good/fair/poor/bad) then act as prototypes in this transformed space, and SCMA learns to map video embeddings toward the appropriate quality region, strengthening the model’s ability to distinguish fine-grained quality differences rather than only semantic differences.

---

> > ### Author Response · Authors · 2025-11-19
> >
> > ### **(3) Empirical Evidence.**
> > This perspective is consistent with our ablation results:
> >
> > #### **1. Comparison with Other Adaptation Strategies (Table 3).**
> > Under an identical CLIP backbone and training data, we compare full fine-tuning, CLIP-Adapter, LoRA, and our SCMA. With only 0.14M trainable parameters, SCMA achieves the highest SROCC/PLCC among all these strategies. This shows that SCMA provides a more effective and significantly more parameter-efficient adaptation mechanism than existing fine-tuning approaches on the same VLMs backbone.
> >
> > #### **2. Structural Ablations of SCMA (Table 4 & Figure 5).**
> > We also perform structural ablations to analyze the effect of each component in SCMA. First, adapting only the visual branch or only the textual branch brings only limited gains over the frozen-VLMs baseline, which indicates that quality alignment needs to involve both modalities. Second, adapting both branches with separate (unshared) adapters yields a larger improvement, showing that jointly tuning visual and textual features is beneficial. Third, when we enforce weight sharing across the two modalities, performance improves again, which suggests that a single common adaptation is more effective than two independent ones. Finally, after adding inter-layer sharing (E-SCMA) and increasing the number of E-SCMA layers, we observe consistent and monotonic improvements. This progression clearly shows that the specific design of SCMA, namely cross-modal and cross-layer shared adaptation, is crucial and yields a stronger and more stable quality-aware representation than naive branch-specific or layer-specific adapters.
> >
> > #### **3. Feature-Space Visualization (Figure 8).**
> > t-SNE plots show that, without SCMA + prompts, visual and textual embeddings are entangled and different quality levels are not clearly separated. After applying SCMA and five-level prompts, the two modalities become better aligned and different quality levels form more compact and ordered clusters, consistent with the view that SCMA reshapes the CLIP joint space into a more quality-discriminative manifold.
> >
> > #### **4. Attention Maps (Figure 10).**
> > We visualize attention maps by computing patch-level similarities between video frames and a guiding quality prompt for four adaptation strategies on the same backbone: full fine-tuning, CLIP-Adapter, LoRA, and our Q-CLIP. The heatmaps show that full fine-tuning often scatters attention over non-critical areas or concentrates mainly on semantic objects that are not informative for quality assessment. CLIP-Adapter and LoRA shift more attention to non-semantic regions than full fine-tuning, but they still fail to cover many regions that actually carry quality-relevant characteristics, and thus do not capture the overall quality information sufficiently. In contrast, Q-CLIP produces attention distributions that are clearly more aligned with quality perception, with stronger focus on regions that are relevant to visual quality rather than purely on semantic saliency. These visualizations provide direct evidence that SCMA, together with the quality prompts, guides the VLMs to attend to quality-relevant content more comprehensively.

---

> ### Author Response · Authors · 2025-11-19
> **Question 3: On the Categorization of Q-Align**
>
> ## **Question 3: On the Categorization of Q-Align**
>
> ## **Answer：**
> **In this paper, we use the term “VLMs” to refer to CLIP-type visual-language models, which typically employ a dual-encoder architecture and are trained through contrastive learning.**
> We agree that Q-Align supports both visual and textual input, so classifying it as a VLMs method from this perspective also seems reasonable. However, we would like to emphasize that this paper categorizes models based on differences in their core architecture, not on their input types. **Q-CLIP and Q-Align exhibit fundamental differences in their core architecture and operational principles:**
>
> - **Q-Align explicitly positions itself as “teaching LMMs for visual scoring via discrete text-defined levels” and operates under the original LMM structure**, where visual tokens are fed into a multimodal Large Language Models (LLMs) that performs the core reasoning and scoring. In this sense, we regard Q-Align as an LLMs-based quality assessment method.
>
>
> - **In contrast, Q-CLIP is built strictly on a CLIP-type, contrastively trained VLMs**: we use only the image encoder and text encoder of a CLIP-type model, keep them entirely frozen, and introduce a single Shared Cross-Modal Adapter (SCMA, 0.14M parameters) as the only trainable module that jointly adapts both modalities for VQA.
>
> Therefore, we classify Q-Align under the “LLMs-based” category, while Q-CLIP falls under VLMs. To avoid confusion, we will add specific notes in the revised version.

---

> > ### Author Response · Authors · 2025-11-27
> >
> > ### **Additional evidence for our use of “VLMs”.**
> >
> > To further clarify the definition of VLMs, we have compiled a substantial set of recent works [1-18] as supporting evidence. **In these papers, the term vision-language models (VLMs) is consistently used to refer to CLIP-type, contrastively trained dual-encoder architectures. In other words, our paper follows an established convention in the community rather than introducing a non-standard definition: “VLMs” typically denotes CLIP-type contrastive models.**
> >
> > Although Q-Align involves both visual and textual modalities, it fundamentally relies on the generative capability of a Large Language Model (LLMs) to produce discrete rating levels. The Q-Align paper explicitly positions itself as a method based on LMMs, rather than a CLIP-type VLMs approach.
> >
> > [1] Wasim, Syed Talal, et al. "Vita-clip: Video and text adaptive clip via multimodal prompting." Proceedings of the IEEE/CVF Conference on Computer Vision and Pattern Recognition (CVPR). 2023.
> >
> > [2] Lu, Yuning, et al. "Prompt distribution learning." Proceedings of the IEEE/CVF Conference on Computer Vision and Pattern Recognition (CVPR). 2022.
> >
> > [3] Menon, Sachit, and Carl Vondrick. "Visual Classification via Description from Large Language Models." The Eleventh International Conference on Learning Representations (ICLR).
> >
> > [4] Shu, Manli, et al. "Test-time prompt tuning for zero-shot generalization in vision-language models." Advances in Neural Information Processing Systems (NIPS) 35 (2022): 14274-14289.
> >
> > [5] Wu, Wenhao, Zhun Sun, and Wanli Ouyang. "Revisiting classifier: Transferring vision-language models for video recognition." Proceedings of the AAAI Conference on Artificial Intelligence (AAAI). Vol. 37. No. 3. 2023.
> >
> > [6] Rasheed, Hanoona, et al. "Fine-tuned clip models are efficient video learners." Proceedings of the IEEE/CVF Conference on Computer Vision and Pattern Recognition (CVPR). 2023.
> >
> > [7] Khattak, Muhammad Uzair, et al. "Maple: Multi-modal prompt learning." Proceedings of the IEEE/CVF Conference on Computer Vision and Pattern Recognition (CVPR). 2023.
> >
> > [8] Lee, Dongjun, et al. "Read-only prompt optimization for vision-language few-shot learning." Proceedings of the IEEE/CVF International Conference on Computer Vision (ICCV). 2023.
> >
> > [9] Yao, Hantao, Rui Zhang, and Changsheng Xu. "Visual-language prompt tuning with knowledge-guided context optimization." Proceedings of the IEEE/CVF Conference on Computer Vision and Pattern Recognition (CVPR). 2023.
> >
> > [10] Singha, Mainak, et al. "Ad-clip: Adapting domains in prompt space using clip." Proceedings of the IEEE/CVF International Conference on Computer Vision (ICCV). 2023.
> >
> > [11] Udandarao, Vishaal, Ankush Gupta, and Samuel Albanie. "Sus-x: Training-free name-only transfer of vision-language models." Proceedings of the IEEE/CVF International Conference on Computer Vision (ICCV). 2023.
> >
> > [12] Chen, Guangyi, et al. "PLOT: Prompt Learning with Optimal Transport for Vision-Language Models." The Eleventh International Conference on Learning Representations (ICLR).
> >
> > [13] Ren, Shuhuai, et al. "Prompt pre-training with twenty-thousand classes for open-vocabulary visual recognition." Advances in Neural Information Processing Systems (NIPS) 36 (2023): 12569-12588.
> >
> > [14] Gao, Peng, et al. "Clip-adapter: Better vision-language models with feature adapters." International Journal of Computer Vision (IJCV) 132.2 (2024): 581-595.
> >
> > [15] Xu, Chen, et al. "Progressive visual prompt learning with contrastive feature re-formation." International Journal of Computer Vision (IJCV) 133.2 (2025): 511-526.
> >
> > [16] Abdul Samadh, Jameel, et al. "Align your prompts: Test-time prompting with distribution alignment for zero-shot generalization." Advances in Neural Information Processing Systems (NIPS) 36 (2023): 80396-80413.
> >
> > [17] Zhu, Yuhan, et al. "Awt: Transferring vision-language models via augmentation, weighting, and transportation." Advances in Neural Information Processing Systems (NIPS) 37 (2024): 25561-25591.
> >
> > [18] Khattak, Muhammad Uzair, et al. "Learning to prompt with text only supervision for vision-language models." Proceedings of the AAAI Conference on Artificial Intelligence (AAAI). Vol. 39. No. 4. 2025.

---

> ### Author Response · Authors · 2025-11-19
> **Question 4&5: Temporal Information Modeling**
>
> ## **Question 4&5: Temporal Information Modeling**
>
> ## **Answer：**
> Given that Reviewer's Questions 4 and 5 both pertain to temporal information modeling, we will address these two questions together.
>
>
> ### **(1) Q-CLIP Incorporates Temporal Information.**
> We acknowledge that temporal information in videos is crucial in VQA. However, we disagree with the reviewers' assertion that using average pooling operations disregards temporal information. We will elaborate on this point in detail below：
>
> #### **1. Evidence that Q-CLIP Incorporates Temporal Information.**
> We evaluate our model across multiple datasets that are not dominated solely by static artifacts, but contain substantial real-world temporal artifacts (e.g., judder, flickering, motion stutter). **Q-CLIP achieves top performance across all six datasets, significantly outperforming methods that explicitly model temporal information** (such as FAST-VQA, MBVQA, and DOVER, **shown in Table 1&2 of our paper**). If Q-CLIP only considered frame-level features, its performance should have been poor on these temporal noise datasets, but **the actual results clearly contradict this expectation**.
>
> #### **2. How Temporal Quality is Captured in Q-CLIP.**
> - Q-CLIP is based on multiple frames rather than a single frame. Our frame-difference-based sampling strategy explicitly considers the temporal information of video by **calculating frame differences to select those frames that best represent the motion information in the video**. Therefore, even though the final aggregation process is simple, temporal information is still incorporated into the Q-CLIP model through the frame selection mechanism.
>
> - Our backbone is a video-tuned VLM, pre-trained on large-scale video data with mean-pooled frame embeddings. In other words, **the backbone has already learned to encode motion patterns and temporal consistency into its per-frame representations under a mean-pooling objective.** Adding complex temporal modeling modules to the model does not enhance temporal sensitivity but instead disrupts the pretrained representation space. Our ablation experiments (Figures 6 & 7) further validate this conclusion: whether on a single dataset or in cross-dataset scenarios, Transformer and Mamba models that explicitly model frame features consistently underperform the mean pooling approach. If mean pooling were incapable of modeling temporal information, its performance should be significantly inferior to Transformer and Mamba. However, the opposite holds true. This demonstrates that mean pooling effectively models temporal relationships.
>
>
> ### **(2) On the “all frames are high-quality but inter-frame inconsistent” Example.**
>
> We believe the reviewer's perspective on this issue may be based on a misunderstanding. We feel it necessary to clarify one point first: **the fact that “every frame is a high-quality image, but the content varies significantly between frames” does not necessarily imply that the video is low quality.** In many real-world scenarios, “every frame is a high-quality image but consecutive frames differ a lot” is exactly what a high-quality video looks like: for instance, PPT lectures or screen-captured talks where each slide is clean but completely different from the previous one, or professionally edited videos and vlogs with frequent hard cuts between scenes. In these cases, large inter-frame changes reflect content editing, not temporal artifacts such as judder, flicker, or frame corruption, and human viewers typically still assign high subjective quality scores. In such settings, treating the video as high-quality is not a “mistake” but aligned with human perception.
>
> More importantly, the LSVQ dataset we use for training and evaluation explicitly contains both types of videos the reviewer mentions:
> - videos where frames are individually high quality but content varies strongly (slides, cuts, etc.).
> - videos with genuine temporal distortions (camera shake, irregular frame rates, flicker, etc.) that humans rate as low quality.
> The MOS labels on LSVQ already integrate these nuances. If Q-CLIP systematically misjudged videos with temporal issues as high-quality simply because their frames look good, its SROCC/PLCC on LSVQ would drop significantly. Instead, we achieve state-of-the-art results, which indicates that, in practice, Q-CLIP is not systematically confused by such cases and can effectively model temporal quality factors present in the dataset.

---

> > ### Author Response · Authors · 2025-11-19
> >
> > ### **(3) Suggestion that “fusion should occur at a shallower level”.**
> > We fully understand and agree with the reviewers' perspective: performing temporal modeling at earlier stages that preserve more pixel/local information theoretically facilitates the detection of extremely subtle inter-frame inconsistencies. From a purely modeling capability standpoint, this is reasonable. However, the design choices for Q-CLIP represent the optimal solution based on a trade-off between the architectural characteristics of CLIP and efficiency objectives.
> > If we were to follow the reviewer’s suggestion and perform temporal fusion at an earlier stage where more pixel-level information is preserved, we would essentially need to convert the CLIP encoder into a genuine spatio-temporal encoder. This would:
> > - substantially increase the number of trainable parameters and computational cost.
> > - pose a higher risk of disrupting the pre-trained semantic space of CLIP.
> >
> > which would be at odds with our core goal of highly parameter-efficient adaptation of VLMs. In summary, earlier-level, pixel-oriented temporal modeling represents a valuable avenue for future expansion. We will explicitly discuss this in our paper and designate it as future work.

---

### Official Review · Reviewer_4auA · 2025-11-04

**Soundness:** 2
**Presentation:** 2
**Contribution:** 2
**Rating:** 2
**Confidence:** 4

**Summary:**

This manuscript introduces a No-Reference Video Quality Assessment framework that is built upon a VLM. To overcome the high computational cost of pretraining and the mismatch between semantic-focused pretraining and the multi-factorial nature of perceptual quality, Q-CLIP proposes a parameter-efficient adaptation strategy. The core idea is to freeze a pre-trained, video-tuned VLM backbone  and introduce two lightweight, trainable modules: Shared Cross-Modal Adapter (SCMA) and learnable five-level quality prompts.

The authors conduct extensive experiments on six VQA benchmarks (LSVQ, KoNViD-1k, LIVE-VQC, etc.), demonstrating state-of-the-art performance. The key claims are SOTA accuracy achieved with exceptionally low finetuning cost compared to both traditional VQA methods and other VLM-based adaptation techniques.

**Strengths:**

- The most significant contribution is the method's efficiency. Achieving SOTA results by only finetuning 0.14M parameters is highly compelling.
- The method consistently achieves SOTA or competitive performance across a wide range of intra-dataset and cross-dataset benchmarks. The performance gains over other VLM-utilizing methods like CLIPVQA are notable.
- The systematic study of frame sampling strategies is a useful, practical contribution. The finding that frame-difference-based (motion-aware) sampling improves cross-dataset generalization is insightful.

**Weaknesses:**

- There are lack of explicit temporal modeling in Q-CLIP. The ablation study on frame feature fusion (Fig. 7) shows that simple mean pooling of frame features outperforms explicit temporal modeling architectures like Transformers and Mamba. It looks like to be an aggregation of frame-level quality scores. This raisessignificant doubts about its ability to assess temporal artifacts (e.g., judder, flickering, motion stutter).
- The method's performance heavily relies on a *specific* VLM backbone (Bolya et al., 2025) that was *already* "pre-tuned on video data". Without an ablation using a standard *image-based* CLIP, it is impossible to know if the proposed adapter is a general-purpose VQA adapter or just a small finetuning layer for an already-powerful video model.
- The paper's best-performing sampling strategy, "Q-CLIP-Mixed" (Table 1) , is never defined in the manuscript or Appendix E . This omission makes the top-reported result in Table 1 irreproducible.

**Questions:**

1. How do the authors expect Q-CLIP to handle temporal-specific distortions like motion judder, flickering, or frame drops, which are critical in VQA? Does this not fundamentally limit the method's applicability to primarily spatial, frame-level artifacts?
2. To clarify the novelty of the SCMA and prompts, could the authors please provide ablation results using a standard, *image-based* CLIP backbone (e.g., the original ViT-B/16 from Radford et al., 2021)?
3. Appendix D.2  describes a complex, dataset-specific finetuning process. Does a single, unified strategy (e.g., training the full 0.14M SCMA) not generalize? How sensitive is the model to these specific per-dataset choices, and does this not contradict the paper's central theme of a simple, general adaptation?
4. The "Q-CLIP-Mixed" sampling strategy achieves the best results in Table 1. Could the authors please provide a precise definition of this strategy? Which methods from Appendix E are combined, and in what proportions?

---

> ### Author Response · Authors · 2025-11-17
> **Question 1: Temporal Modeling**
>
> ### We sincerely appreciate the time and effort you have devoted to reviewing our manuscript and are grateful for your insightful comments. Meanwhile, We would like to address the concerns raised, as detailed below:
>
> ## **Question 1: Temporal Modeling**
>
> ## **Answer：**
> We acknowledge that temporal artifacts are an important factor influencing VQA. However, we believe the concern that **the absence of explicit temporal modeling fundamentally limits Q-CLIP to frame-level artifacts arises from a misunderstanding**. This conclusion is neither supported by our empirical experiments nor consistent with prior VLMs-based video research findings. **On the contrary, mean pooling of video frame features has been thoroughly validated as an efficient strategy.** We elaborate below:
>
> ### **(1) Evidence that Q-CLIP Does Handle Temporal Distortions.**
> We evaluate our model across multiple datasets that are not dominated solely by static artifacts, but contain substantial real-world temporal artifacts (e.g., judder, flickering, motion stutter). **Q-CLIP achieves top performance across all six datasets, significantly outperforming methods that explicitly model temporal information** (such as FAST-VQA, MBVQA, and DOVER, **shown in Table 1&2 of our paper**). If Q-CLIP only considered frame-level artifacts, its performance should have been poor on these temporal noise datasets, but **the actual results clearly contradict this expectation**.
>
>
> ### **(2) “No Explicit Temporal Modeling” ≠ “Temporal Blindness”.**
> The reviewer’s argument implicitly assumes that temporal distortions can only be captured by a dedicated sequence model (Transformer/Mamba) on top of frame embeddings. This assumption does not hold in our setting:
>
> - Q-CLIP is based on multiple frames rather than a single frame. Our frame-difference-based sampling strategy explicitly considers the temporal information of video by **calculating frame differences to select those frames that best represent the motion information in the video**. Therefore, even though the final aggregation process is simple, temporal information is still incorporated into the Q-CLIP model through the frame selection mechanism.
> - Our backbone is a video-tuned VLM, pre-trained on large-scale video data with mean-pooled frame embeddings. In other words, **the backbone has already learned to encode motion patterns and temporal consistency into its per-frame representations under a mean-pooling objective.** Adding complex temporal modeling modules to the model does not enhance temporal sensitivity but instead disrupts the pretrained representation space. **Our ablation experiments (Figure 6&7) demonstrate precisely this: both Transformer and Mamba head models consistently underperform simple mean pooling, whether on single datasets or cross-dataset scenarios.**
>
>
> Therefore, Q-CLIP still effectively utilizes temporal information through multi-frame input, motion-aware sampling, and a pre-trained video backbone whose frame embeddings have encoded temporal cues.
>
> ### **(3) Prior VLMs Works Also Reveal Similar Results.**
>
> Previous studies have demonstrated that **for VLMs backbone networks, simple pooling of frame embeddings constitutes an exceptionally powerful temporal aggregation strategy, outperforming even more complex temporal modules.** For example, **CLIP4CLIP** [1] indicates that in video-to-text retrieval tasks, zero-parameter mean pooling outperforms LSTM/Transformer-based temporal modeling on small datasets. Therefore, when training data is limited, adding extra temporal parameters should be avoided. This further validates the soundness and effectiveness of our aggregation strategy employing mean pooling.
>
> ### References:
> [1] Luo, Huaishao, et al. "Clip4clip: An empirical study of clip for end to end video clip retrieval and captioning." Neurocomputing 508 (2022): 293-304.

---

> ### Author Response · Authors · 2025-11-17
> **Question 2: Backbone & Fine-Tuning Ablation**
>
> ## **Question 2: Backbone & Fine-Tuning Ablation**
>
> ## **Answer：**
> We address this with two pieces of evidence: **(1) new experiments with standard image-based CLIP backbones**, and **(2) a comparison of different fine-tuning strategies on the same backbone**.
>
> ### **(1) New Experiments with Standard Image-Based CLIP Backbones.**
> As requested, **we have added experiments on three standard image-based CLIP architectures: ViT-B/16, ViT-B/32, and ViT-L/14** [2]. For each architecture, we maintain the overall Q-CLIP framework unchanged, replacing only the text and visual encoders. We also follow the same experimental setup. We pre-train the proposed Q-CLIP on LSVQ and conduct intra-dataset testing on $LSVQ_{test}$ and $LSVQ_{1080p}$. Additionally, cross-dataset testing performed on KoNViD-1k and LIVE-VQC. Detailed results are shown in the table below:
>
>
> | Methods       | LSVQ$_{test}$ | LSVQ$_{1080p}$ |  KoNViD-1K  |   LIVE-VQC  |
> |---------------|:-------------:|:--------------:|:-----------:|:-----------:|
> | Q-Align       |  0.883/0.882  |   0.797/0.830  | 0.865/0.877 |     -/-     |
> | PTM-VQA       |  0.855/0.864  |   0.736/0.782  | 0.824/0.830 | 0.785/0.737 |
> | CLiF-VQA      |  0.886/0.887  |   0.790/0.832  | 0.877/0.874 | 0.834/0.855 |
> | CLIPVQA       |  0.881/0.883  |   0.782/0.827  | 0.864/0.887 | 0.781/0.871 |
> | **CLIP-B/16** |  0.885/**0.887**  |   **0.799**/**0.842**  | **0.881**/0.878 | 0.803/0.843 |
> | **CLIP-B/32** |  **0.887**/**0.887**  |   **0.797**/**0.843** | **0.879**/0.879 | 0.799/0.847 |
> | **CLIP-L/14** |  **0.893**/**0.894**  |   **0.814**/**0.852**  | **0.885**/**0.892** | 0.811/0.855 |
> | **Q-CLIP**    |  **0.899**/**0.900**  |   **0.823**/**0.866** | **0.896**/**0.901** | 0.826/0.867 |
>
> It can be seen that when using three image-based CLIP backbones (CLIP-B/16, CLIP-B/32, CLIP-L/14), the model achieves performance comparable to the current state-of-the-art. **These results demonstrate that the proposed adapter and prompt design can be effectively transferred to different CLIP backbone models (purely image-trained models), rather than being limited to specific video models.** Additionally, **when using video-tuned backbones, Q-CLIP achieves a slight performance improvement.** This indicates that the video-tuned backbone employed by Q-CLIP does contribute to performance gains, but it is not the sole reason for our method's strong performance.
>
> ### **(2) Evidence from Fine-Tuning Strategies on Same Backbone.**
> Beyond changing the backbone, the main paper already contains an ablation (**Table 3**) that isolates the effect of the adaptation strategy given exactly the same video-tuned backbone. **In Table 3, we compare: full fine-tuning of the backbone, CLIP-Adapter, LoRA, and our proposed SCMA, under the same backbone and training setup.** The results show that SCMA achieves higher SROCC/PLCC than full fine-tuning, CLIP-Adapter, and LoRA. This comparison is particularly informative for the reviewer’s concern: since the backbone is strictly identical in all four settings, the performance differences can only come from how the backbone is adapted. **The fact that SCMA significantly outperforms other mainstream fine-tuning strategies demonstrates that Q-CLIP's advantages are not solely attributable to the robust performance of its backbone network**.
>
>
> ### References:
> [2] Radford, Alec, et al. "Learning transferable visual models from natural language supervision." International Conference on Machine Learning. PMLR, 2021.

---

> > ### Author Response · Authors · 2025-11-29
> >
> > Additionally, following reviewer HNXL's suggestion, we further conducted experiments comparing Q-CLIP with CoOp and VPT under the same CLIP backbone, as shown in the table below. As can be seen, **CoOp performs poorly (0.763/0.764)**, which is expected since it only optimizes prompts while keeping all backbone parameters frozen, and thus cannot sufficiently adapt the representations to fine-grained video quality. **VPT achieves acceptable results (0.823/0.820) by introducing visual prompt tokens, but it is still clearly behind CLIP-Adapter/LoRA and our method.**
> >
> > By contrast, Q-CLIP achieves 0.897/0.895, outperforming CoOp, VPT, full fine-tuning, CLIP-Adapter, and LoRA on the same backbone. **These results further validate the effectiveness of the method we proposed.**
> >
> > | Methods          |    SROCC/PLCC   |
> > |------------------|:---------------:|
> > | CoOp             |   0.763/0.764   |
> > | VPT              |   0.823/0.820   |
> > | Full fine-tuning |   0.816/0.811   |
> > | CLIP-Adapter     |   0.881/0.884   |
> > | LoRA             |   0.883/0.883   |
> > | **Q-CLIP**       | **0.897/0.895** |

---

> ### Author Response · Authors · 2025-11-17
> **Question 3: Clearance of Fine-Tuning & Generalizability**
>
> ## **Question 3: Clearance of Fine-Tuning & Generalizability**
>
> ## **Answer：**
> Actually, our intention there was not to contradict the simplicity and generality of Q-CLIP, but rather to report some minor, dataset-specific refinements that we found could squeeze out a small amount of extra performance on very small datasets.
>
> ### **(1) Single Unified Fine-Tuning Strategy Already Achieves SOTA.**
> In response to this comment, we have conducted additional experiments where we use one unified fine-tuning configuration for all small datasets (LIVE-VQC, KoNViD-1k, CVD2014, YouTube-UGC, LIVE-Qualcomm). Concretely, we fine-tune the entire 0.14M-parameter SCMA module with a single learning rate and identical training schedule for all these datasets, without any dataset-specific choices.
> The results show that, under this unified setting, Q-CLIP still achieves state-of-the-art performance on all small datasets, and remains clearly stronger than prior VQA methods.
>
> | Methods |   LIVE-VQC  |  KoNViD-1K  | YouTube-UGC |   CVD2014   | LIVE-Qualcomm |
> |---------|:-----------:|:-----------:|:-----------:|:-----------:|:-------------:|
> | SCMA    | 0.879/0.900 | 0.913/0.920 | 0.911/0.911 | 0.898/0.906 |  0.844/0.881  |
> | Q-CLIP  | 0.881/0.901 | 0.915/0.920 | 0.911/0.911 | 0.897/0.907 |  0.846/0.884  |
>
> ### **(2) Dataset-Specific Strategies Provide Only Marginal Gains.**
> The per-dataset strategies described in Appendix D.2 were introduced purely as a practical refinement: in some cases they bring a slight additional improvement over the unified setting (typically a very small increase in SROCC/PLCC). In other words, they are optional tuning tips, not a requirement for Q-CLIP to work. The overall performance of Q-CLIP is not highly sensitive to these choices, as long as the SCMA module is properly fine-tuned.

---

> ### Author Response · Authors · 2025-11-17
> **Question 4：Definition of “Q-CLIP-Mixed” Sampling Strategy**
>
> ## **Question 4：Definition of “Q-CLIP-Mixed” Sampling Strategy**
>
> ## **Answer：**
> We apologize for the lack of a precise definition in the current draft and will clarify it in the revision. In our implementation, **Q-CLIP-Mixed is a mixture of all six frame sampling strategies described in Appendix.** E. It is used in both training and inference as follows:
>
> - **Training.** For each video in each iteration, we randomly select one of the six sampling strategies with equal probability (i.e., a uniform 1/6 chance for each strategy) to extract frame subsets. In this way, the model is exposed to diverse sampling patterns and learns to be robust to how frames are selected, rather than overfitting to a single fixed strategy.
> - **Inference.** We simultaneously apply all six sampling strategies to the video, then use the average of all sampled predictions as the final prediction. Therefore, Q-CLIP-Mixed can be regarded as a simple ensemble method that combines the six sampling schemes with equal weights.

---

### Author Response · Authors · 2025-11-20

We would like to sincerely thank all reviewers (R1: 4auA, R2: AvtS, R3: HNXL, R4: rJe3) for their careful comments and constructive feedback. We are greatly encouraged that our approach is **efficient** (R1, R2, R3, R4), **effective** (R1, R2, R3, R4), **sound** (R3) and **well-motivated** (R3). Our sampling strategy is **practical** (R1) and **insightful** (R1, R3). Our experiments are **adequate** (R1, R3, R4) and our paper is **well-written** (R2, R3, R4).

**We have carefully reviewed each comment and provided detailed point-by-point responses**, including necessary supplementary analysis, clarifications and discussions. We hope these responses address the reviewers' concerns and further clarify the contributions and significance of this study. **We would be very grateful if the reviewers could consider our answers, and we are happy to provide any further explanations if some aspects remain unclear.**

---

> ### Author Response · Authors · 2025-11-27
>
> In addition, we have incorporated the reviewer’s constructive comments and the suggested additional experiments into the revised manuscript, and we have highlighted all modified and newly added content in blue.

---

### Meta-Review · Area_Chair_EaZ5 · 2026-01-09

**Summary:**

This paper proposes Q-CLIP, a parameter-efficient VLM-based framework for video quality assessment using a shared cross-modal adapter and learnable quality prompts. Reviewer discussion mainly focused on the actual novelty beyond existing CLIP-based adaptations, the lack of explicit temporal modeling, and whether the reported gains primarily stem from a strong pretrained backbone rather than the proposed method itself.

**Reviewer Concerns:**

While the authors provided extensive additional experiments and clarifications, several core concerns remain only partially resolved. In particular, doubts persist regarding the method’s conceptual novelty relative to prior VLM-based VQA approaches, the limited and indirect handling of temporal distortions, and the reliance on a strong video-tuned CLIP backbone. These issues affect the paper’s significance and positioning rather than its empirical performance.

**Reviewer Scores:**

Reviewer 4auA: would likely remain reject.
Reviewer AvtS: would likely remain below acceptance threshold.
Reviewer HNXL: might maintain a marginal accept but expressed uncertainty.
Reviewer rJe3: increased score but still raised concerns about novelty and positioning.

---

### Decision · Program_Chairs · 2026-01-26

Reject